# Parallel but distinct adaptive routes in the budding and fission yeasts after 10,000 generations of experimental evolution

Experimental evolution has been a useful tool for investigating long-term temporal evolutionary dynamics and molecular mechanisms underlying adaptation. However, extracting fundamental principles and predictive features of evolutionary outcomes from these datasets remains challenging. Here we sought to circumvent these challenges by comparing distant yeast species that share several evolutionary features but differ in evolutionary history and genome architecture, that is *Saccharomyces cerevisiae* and *Schizosaccharomyces pombe*. We evolved ten populations of the fission yeast for 10,000 generations in the same conditions as a pre-existing budding yeast dataset, allowing us to observe repeatable evolutionary outcomes within species but diverse molecular targets of adaptation across species. The most frequent route of adaptation was through changes in carbon flux metabolism, which was previously unseen in *S. cerevisiae* evolved populations, but similar evolutionary paths have been observed in wild populations. This suggests that parallelism is pervasive and that mechanisms of adaptation can be shared among closely related or distant species. Despite similar gene content and identical environments, recurrent adaptation across *S. pombe* populations involved different genes than in *S. cerevisiae* and was detectable mostly at the transcriptomic level. This indicates that *trans*-regulatory effects and contingency may contribute to differences in evolutionary outcomes between these species.

Long-term experimental evolution has highlighted the predictability of high-level features of evolution, such as fitness trajectories[1–6]. Although fitness gains follow a reproducible pattern of declining adaptability, the exact mutations or the sequence in which they occur are typically unpredictable. Nevertheless, experimental evolution done in large replication with the budding yeast *Saccharomyces cerevisiae* has revealed extensive parallelism at the gene or pathway level[3,7,8], even across very different strains[9]. These reproducible patterns during experimental evolution are also seen in different organisms. For instance, during *Escherichia coli* long-term experimental evolution (LTEE), half of the populations evolved a mutator phenotype and quasi-stable lineage co-existence emerged in 75% of the populations[4]. Further, substantial parallelism at the gene level can also be observed throughout 12 populations over 60,000 generations[4].

Unsurprisingly, the molecular targets of selection in these two species differ substantially[3,6]. Likewise, comparing microbial laboratory evolution with experimental evolution in *Caenorhabditis elegans* and *Drosophila melanogaster* does not reveal universal targets of adaptation[10,11]. Notwithstanding their vastly different gene content and cellular form, differences in adaptation are further explained by the different selection environments (the LTEE was done in DM25 minimal medium, whereas yeast experiments were in rich medium).

✉e-mail: alex.nguyenba@utoronto.ca

Another interesting hypothesis for these differences relates to the role of evolutionary contingency in shaping the dynamics and targets of selection. Contingency is the product of past evolution, whereby epistasis and genome structure strongly influence future adaptive potential[1,12,13]. Contingency has been explored in the context of strains with different gene deletions[13–16] and has even been observed in experimental evolution. For instance, across the 12 LTEE *E. coli* populations, 1 population developed the ability to grow aerobically on citrate—a rare phenotype in this species—following a genome rearrangement. This phenotype is more readily evolvable within that population after this rearrangement[17]. Further, although LTEE populations evolved mutator phenotypes or stable lineage co-existence, these phenomena are not observed in *S. cerevisiae* laboratory evolution experiments in standard media, despite being capable of occurring through genetic engineering[18].

Although studying evolution using very distinct organisms has provided insights into features of evolutionary dynamics, it has been difficult to discern the effect of life history from other idiosyncrasies on the molecular targets of selection. Would we find the same mutated genes, and a preference for the same molecular mechanisms, if we had evolved a species with substantially similar gene contents to that of *E. coli* or *S. cerevisiae*? Evaluating the impact of differences between such species on quantitative trait evolution could improve our understanding of the role of genetic architecture and life history in trait diversity.

One way to address the influence of these factors on adaptation is through the yeast subphylum. Although it exhibits immense trait diversity and is as genetically diverse as the animal kingdom[19,20], species in the yeast subphylum have comparable number of genes, and share many genes and metabolic pathways such that most species can grow in the same media. As an example, *S. cerevisiae* and *Schizosaccharomyces pombe* share about 4,500 genes (out of approximately 6,000), but are so diverged that their genomes show no synteny[21,22]. Evolutionarily, they have similar spontaneous mutation rates and were both domesticated for their ability to ferment. Some notable differences are that *S. pombe* is naturally a haploid in the wild[23,24], whereas *S. cerevisiae* is usually a diploid[25,26]. This could affect the effectiveness of selection because mutations can be recessive in diploids. Therefore, the extant genome might have 'footprints' of this accumulated genetic load in *S. cerevisiae*[3,27] and this is supported by heterosis in wild strains[28,29]. Further, *S. cerevisiae* is within the clade of yeasts that have undergone the 'whole-genome duplication event', which has profoundly shaped their metabolism due to functional divergence in paralogous genes[30]. Another principal difference is that *S. pombe* cannot survive the loss of its mitochondrial DNA and relies on oxidative phosphorylation for growth, that is, it is petite-negative, whereas *S. cerevisiae* can survive without mitochondrial DNA, that is, it is petite-positive, highlighting a rare ability of budding yeast[31–33]. Thus, different life cycles and evolutionary histories might have led *S. cerevisiae* to produce and process ethanol more efficiently than *S. pombe*[34,35] and could lead to different evolutionary outcomes.

Hints of the effects of contrasting life and evolutionary histories on evolutionary signatures emerge when comparing the molecular processes that are under selection and could potentially allow for the prediction of molecular evolution. For example, *S. cerevisiae* gene content variation across strains is more frequent than in *S. pombe*, and loss-of-function (LOF) mutations, such as nonsense mutations and frameshift deletions, tend to be fixed recurrently across populations[36]. Previous genomic analyses also showed that 43% of *S. pombe* genes have introns compared with 5% in *S. cerevisiae*, and upstream intergenic regions are larger in the former[37]. Annotated coding regions span around 70% of the *S. cerevisiae* genome compared with 60% for *S. pombe*[38–40]. This might imply that *S. pombe* would preferably adapt through regulatory changes as mutations in non-coding regions of *S. pombe* are known to be involved in transcription regulation and

should affect trait evolution and adaptation[41]. Budding yeast, on the other hand, may do so through copy-number variation and gene inactivation. Thus, comparing the evolutionary processes and targets of adaptation in *S. cerevisiae* and *S. pombe* through experimental evolution could help highlight factors explaining differences in evolutionary outcomes across and within species.

To address the gap in understanding how species with similar gene content, but different life histories, might evolve in the same environment, we undertook laboratory evolution, where we founded about 16 populations each of five different budding yeast species, as well as the *S. pombe* fission yeast. Here we describe the dynamics and molecular evolution of the fission yeast populations over the first 10,000 generations. Fission yeast is the outgroup of this set of species, and serves as a useful anchor for expectations within our chosen fungal phylum. We analyse the mutational spectrum, the molecular evolutionary dynamics and identify, through pathway enrichment and whole-transcriptome sequencing, a case of parallel evolution where *S. pombe* populations have rewired their central carbon metabolism to reduce dependency on oxygen, which mirrors similar events in wild *S. pombe* populations and in related *Schizosaccharomyces*[32,35,42,43].

## Results

### Contrasting fitness gain of fission yeast during 10,000 generations of evolution

To compare the evolutionary processes contributing to adaptation and selection targets between *S. pombe* and *S. cerevisiae*, we founded 15 *S. pombe* populations and performed experimental evolution for 10,000 generations, collecting and freezing samples approximately every 70 generations (Fig. 1; Methods). The populations were founded from single colonies of the same *S. pombe* haploid laboratory strain and serially propagated in wells of lidded, unshaken 96-well plates in rich medium with daily dilution. The growth conditions were identical to our previous experimental evolution study, which analysed 10,000 generations of evolution for *S. cerevisiae* populations passaged using liquid handling robotics[3]. Five populations were lost during the evolution due to contamination or mechanical pipetting errors. Because of the plate lidding and the absence of shaking, these conditions correspond to a high-sugar hypoxic environment. Although selection pressure faced by the species may differ, our experiment controls for many of the environmental variables for the comparison of evolutionary outcomes.

To determine whether the populations adapted during evolution, we measured the fitness at the initial and final time points (Eq. 1; Methods). The evolved populations are all fitter than the ancestral strain (Extended Data Fig. 1) as observed in other experimental evolution studies[3,6]. However, the fitness gains of the *S. pombe* lineages after evolution are smaller than those of haploid populations of *S. cerevisiae* in the same environment (Extended Data Fig. 1). This slower rate of fitness gain is not due to *S. pombe* being 'pre-adapted' to this environment, as the ancestor *S. cerevisiae* strain is much fitter than the ancestor *S. pombe* strain used here. We presume that *S. cerevisiae*, having higher glycolytic and fermentation rates, can adapt more readily to the experimental conditions through a more diverse set of stress response pathways from having explored anaerobic niches throughout its evolutionary history[32,34,35,44]. *S. pombe*, on the other hand, is a poor fermenter in anaerobic conditions[32] and has a less robust hypoxic stress (HS) response[45]. Thus, each species may evolve through different adaptation strategies with different associated costs, explaining the differences in adaptation rate.

### Differences in the genome architecture partially explain the different mutation spectra of *S. pombe* and *S. cerevisiae*

To uncover differences in adaptation strategies between the two species and the molecular mechanisms underlying them, we performed whole-population, whole-genome sequencing every 1,000 generations on the *S. pombe* populations. We sequenced successfully cells from

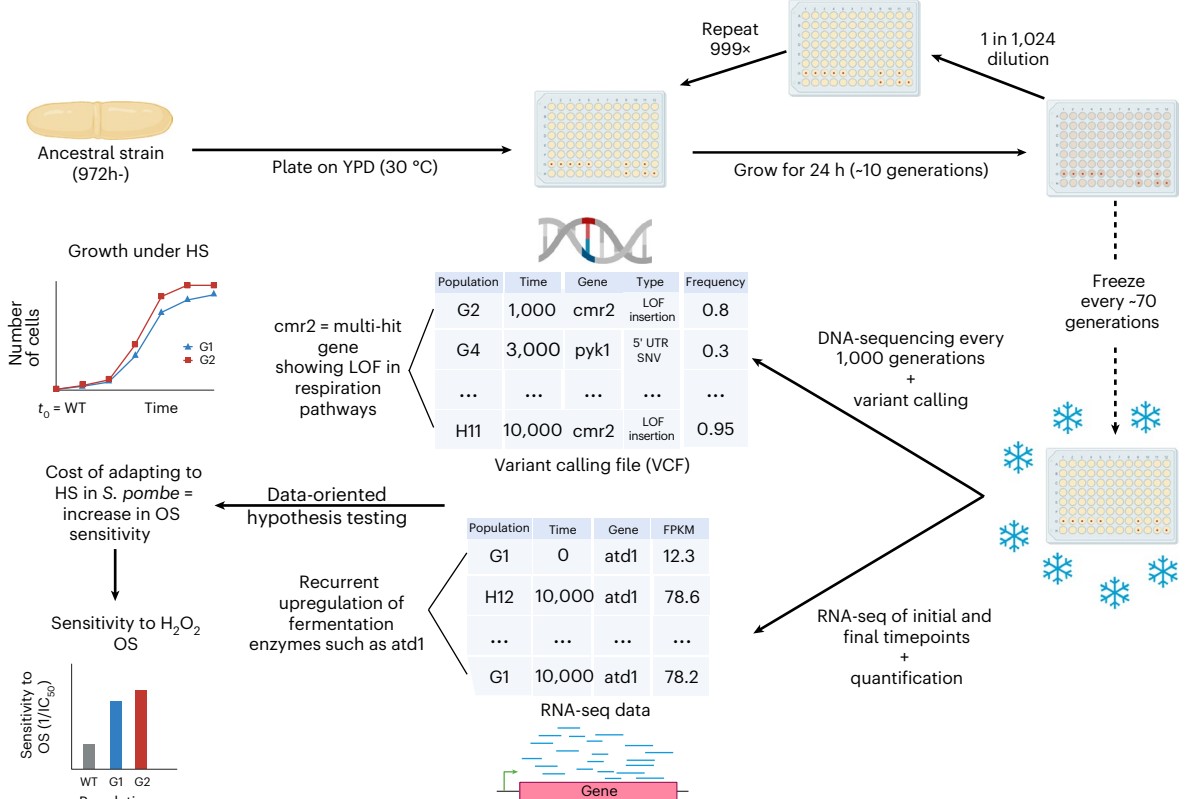

**Fig. 1 | Experimental approach.** We evolved 15 *S. pombe* populations at 30 °C in lidded YPD 96-well plates for 10,000 generations, collecting and freezing samples approximately every 70 generations in 27% glycerol at −80 °C and sequenced them every 1,000 generations. These populations were part of a larger experimental evolution project, which explains why they occupy two rows of the 96-well plates, and were examined regularly for contamination (Methods). Because the experimental conditions yielded a doubling rate of ten generations per day, we performed a 1 in 1,024 serial dilution every day ($1/2^{10}$), which can be done by two consecutive 1:32 dilutions with 96-well plates. This yielded ~100 time points or samples from which we sequenced DNA to identify variants. We also sequenced RNA at the initial (WT) and final time points to quantify gene expression changes. To find targets of selection at the genomic levels, we identified multi-hit genes, that is, genes that were recurrently hit by fixed mutations (frequency ≥75%) that are not synonymous (indels, missense or nonsense) in several populations. At the transcriptomic level, we identified genes that were expressed differentially across several evolved populations compared with WT or with populations not exhibiting a change in a phenotype of interest, for example, sensitivity to OS. These data allowed us to generate hypotheses about adaptation in the ten populations surviving, that is, G1–G5, G9, H9 and H11 and 12, which we tested experimentally. For instance, in the figure, G1 (blue) and G2 (red) both adapt to HS at the cost of becoming more sensitive to OS. Figure created in BioRender; Arnaud, N. https://biorender.com/8pi06vc (2026).

93 out of the 100 stored samples. The remaining seven were presumed lost during storage and were not associated to any specific population or archival date. We applied several filters to the called variants[46,47] while leveraging the time series for consistency to account for variant-calling errors (Methods).

To validate the accuracy of the called variants, we measured the substitution rates and the fixed insertion-to-deletion ratios, in each *S. pombe* population. We measured a fixed insertion-to-deletion ratio ranging from 2.11 to 6.33 (median = 4.43, $\mu$ = 4.32 and $\sigma$ = 1.22; Extended Data Fig. 2), consistent with the insertion preference of *S. pombe*[48]. For the substitution rate, we measured an average of $1.91 \times 10^{-10}$ fixed substitutions per site per generation (median = $1.72 \times 10^{-10}$, $\sigma = 6.76 \times 10^{-11}$), which is similar to results from mutation accumulation experiments in *S. pombe* ($2.00 \times 10^{-10}$ in ref. 48) and within the range of *S. cerevisiae*'s rate estimates ($1.67 \pm 0.04 \times 10^{-10}$ in ref. 49 and $3.30 \pm 0.80 \times 10^{-10}$ in ref. 50).

Although both species accumulate substitutions at a similar rate, their genome architectures differ. Deviations from a null model of evolution in which all loci have the same chance of being hit by substitutions would suggest the presence of selection in certain loci and a preference for specific types of mutation. Thus, we compared the frequency of different variants observed in the *S. pombe* populations with that observed in the experimental evolution of *S. cerevisiae* haploid populations[3].

The fixation of variants in the non-coding regions of *S. pombe* is higher than expected by the null model, suggesting positive selection in non-coding regions or purifying selection in coding regions (Fig. 2). To test whether this reflects a signal of purifying selection in protein-coding genes, we compared the indel rate, that is, indels per kilobase, first stratifying by coding or non-coding and then by whether the indel size was a multiple of three (Extended Data Fig. 3). In protein-coding genes, controlling for increased likelihood of shorter indels, we find that indels that preserve reading frame are enriched compared with those that do not (Extended Data Fig. 3). In contrast, indels in non-coding regions do not exhibit such selective constraints on their size. This is similar to observed polymorphisms in humans and budding yeast[51] and suggests purifying selection in protein-coding genes. This, and the paucity of fixed coding mutations compared with *S. cerevisiae* populations, suggests that the genetic context of *S. pombe* is less amenable to tinkering through LOF mutations. One hypothesis for this may be that the whole-genome duplication event in *S. cerevisiae* provided more genetic redundancy, which can increase evolvability through coding mutations or through rapid changes in functional gene content[30,52]. An additional possibility is that *S. pombe* has more essential genes than *S. cerevisiae* (26.1% versus 17.8%)[53]. Altogether, these results show that differences in the mutation spectra of *S. cerevisiae* and *S. pombe* are explained only partially by their genome architectures.

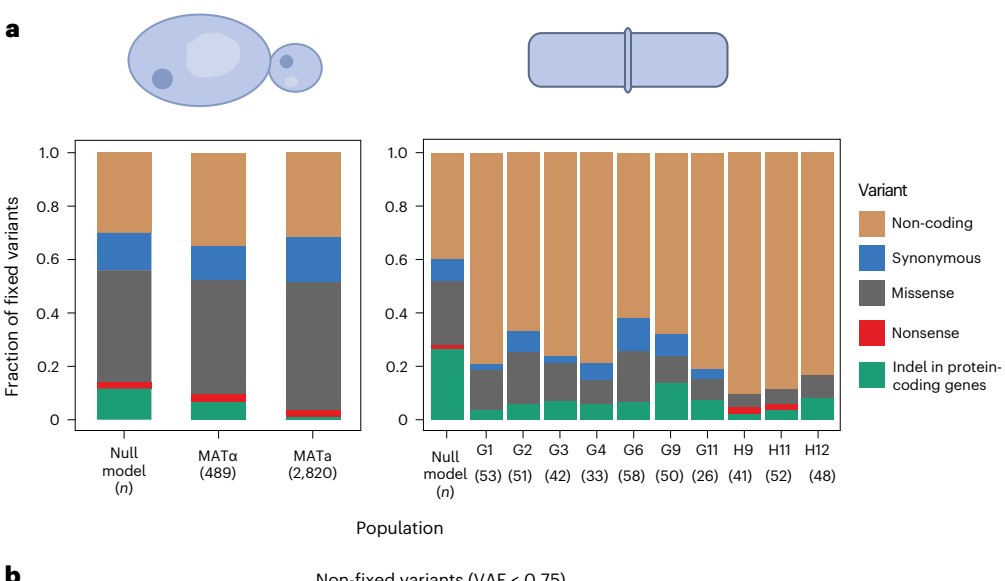

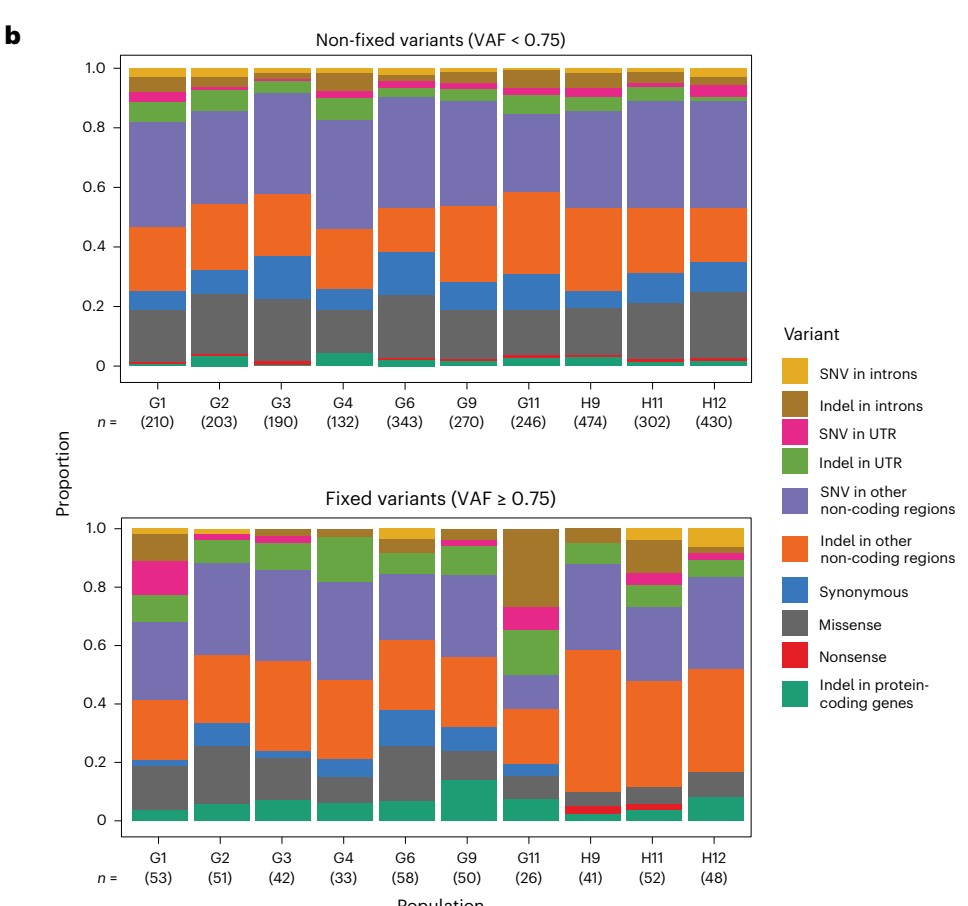

**Fig. 2 | Evolution spectra and fixed mutation biases in budding and fission yeasts. a**, The fixed variant (frequency ≥75%) spectrum in haploid populations of *S. cerevisiae* after 10,000 generations of experimental evolution (left: data from ref. 3) compared with that of *S. pombe* (right). The experimental evolution protocol is the same for both species and is described in ref. 3. The null model is obtained by calculating the expected number of fixed variants in each category based on the species' evolutionary features (equations (2) and (3); Methods). Sample size (*n*) represents the number of fixed mutations used to calculate the proportions. In the Johnson et al.[3] dataset, it represents the total observed in all the haploid populations (7 MATα and 36 MATa). **b**, Variants in each of the ten evolved *S. pombe* populations. Top: non-fixed variants, bottom: fixed variants. VAF, variant allele frequency; SNV, single nucleotide variant; UTR, untranslated region.

## Genomic targets of adaptation in *S. pombe* are genes involved in the response to hypoxic and oxidative stresses

Despite signatures of purifying selection in protein-coding genes, we cannot exclude the presence of loci evolving under positive selection. However, we may expect different targets of adaptation between fission and budding yeasts due to differences in their evolutionary histories and genetic architectures.

Johnson et al.[3] looked for targets of selection using signals of convergent evolution in evolving budding yeast populations. They defined a 'hit' as a fixed protein-altering variant, and a multi-hit gene as having

hits in several populations. They identified 189 genes with hits in at least six populations that were enriched in functions related to the adenine biosynthesis pathway, the mating pathway and negative regulators of the Ras pathway. Using the same approach with a minimum of two out of ten populations in which a gene has a hit, we identified 19 multi-hit genes involved in transcription regulation, protein degradation, autophagy, ion transport, flocculation, stress response, respiration and glycolysis (Supplementary Table 1). These genes had hits in either the cis-regulatory region or the coding region. We note, however, that 11 indels were also in multi-hit genes. These may indicate genetic-level nucleotide convergence or strong mutational biases (see Methods for details on how we rule out sequencing artefacts for these). Multi-hit genes were more recurrent than expected under a null model where variants are fixed at the same rate across all the genes (two-sided Wilcoxon rank sum $P$ value < 0.05; Extended Data Fig. 4; Methods). This signal of positive selection is also consistent with the fixation of the same hit at different times in several populations (Fig. 3, Supplementary Table 1 and Extended Data Fig. 5). Finally, consistent with a different indel bias than in *S. cerevisiae*[3] (Fig. 2 and Supplementary Table 1) and a preference for insertion (Extended Data Fig. 2), we also found adaptation through LOF frameshift insertions (rather than deletions).

Most of the *S. pombe* multi-hit genes are involved in the response to HS and sometimes to oxidative stress (OS) (Supplementary Table 1), which may be due to our unshaken and lidded environment. Adapting to HS through fermentation produces ethanol and acetate, which can hinder mitochondrial membrane integrity and functions, leading to production of reactive oxygen species (ROS)[44,54–58]. Fermentation also yields smaller energy and biomass outputs than respiration[32,34], affecting the synthesis of redox components. Finally, *S. pombe* is sensitive to polypeptones included in the growth medium of the experiment, and the respiratory Coenzyme Q (CoQ10) plays a role in promoting this trait[59,60]. *S. cerevisiae*, on the other hand, is petite-positive with a robust and adapted fermentation pathway, which may explain why such mutations are not typically observed in budding yeast experimental evolution studies.

Further supporting the importance of energy- and ROS-related processes in this environment is the strongest signal of convergent evolution in the lower glycolysis gene *pyk1*, which had fixed mutations in five populations. This gene encodes the enzyme involved in the synthesis of pyruvate, influencing respiration and fermentation as it can be converted into Acetyl-CoA or into acetate and ethanol[35]. In the population H11, the glycolytic flux going towards respiration seems to be further compromised by a fixed LOF mutation in *cmr2* (Fig. 3), which encodes the enzyme converting pyruvate into Acetyl-CoA. The fixation of this LOF mutation might also be related to a benefit in re-orienting the glycolytic flux towards fermentation.

## Most evolved *S. pombe* populations are more sensitive to OS

Populations adapting to HS or to polypeptones may exhibit decreased respiration rates, resulting in reduced energy and biomass production[32]. This could affect the synthesis of the components of the redox metabolism[61,62]. These enzymes and those involved in the respiratory chain rely on the energy produced by glycolysis and the tricarboxylic acid cycle cycle as well as ions like iron and sulfur for their functions[32,61,63,64]. Consistent with this, we observed the presence of hits in the gene *fet4* encoding an iron transporter in several populations. Moreover, one population harbours a LOF insertion in *prr1* (Fig. 3 and Supplementary Table 1)—a gene encoding an important transcription factor involved in the response to OS[65,66]. Altogether, these observations suggest that the evolved populations were more sensitive to OS than wild type (WT) as a result of their adaptation to HS.

To test this, we used hydrogen peroxide ($H_2O_2$) as a source of OS (Methods) and compared the OS sensitivity of the WT and of the ten evolved populations (Fig. 4a). The populations clustered into three groups: three out of ten populations had a similar sensitivity to $H_2O_2$ as

the WT, five were more sensitive than WT and two, including H11—the population that lost the OS response regulator *prr1*, were much more sensitive than the WT strain (Fig. 4b,c and Extended Data Fig. 6). This increased sensitivity to OS after adapting to HS has not been observed in *S. cerevisiae*, which may be explained by contingency, that is, more exposure to anaerobic environments during its evolutionary history and a better redox metabolism[32,34]. However, it is reproducible in most of the evolved *S. pombe* populations, and has likewise been observed in related wild *Schizosaccharomyces japonicus*[32] ('Discussion').

## Transcriptomics analysis reveals many genes involved in the response to HS and downstream effects explaining the sensitivity to OS in several populations

The sensitivity to OS developed after adapting to HS was revealed by convergent evolution at the genomic level. However, identifying transcriptomic differences between more sensitive and WT-like populations could help reveal genes driving this evolutionary outcome and explain how point mutations and indels contribute to the evolution of these traits. Moreover, most of the mutations in the pyruvate kinase (*pyk1*) locus are in the untranslated regions flanking the gene (Extended Data Fig. 7). Hits in these loci could explain the adaptation to HS through changes in transcript levels[67–71]. These questions led us to investigate the role of transcriptomic changes in the responses to HS and OS by performing RNA-seq of the WT ancestor and the final evolved populations and comparing the transcriptomic profiles of populations more sensitive to $H_2O_2$ than WT with those of populations as sensitive as WT (Methods).

Consistent with the need to compensate for biomass and energy deficit under HS due to limited respiration[32], most evolved *S. pombe* populations upregulated genes involved in haem metabolism/transport, the transport of biomolecules, lipid metabolism, fermentation and ion transport (Fig. 5 and Supplementary Table 2). Another way the fission yeast populations could respond to HS is by improving the electron transport chain (ETC) to leverage the limited amount of oxygen available. The upregulation of the haem transporter genes *str3* and *shu1* in many evolved populations supports such a strategy (Supplementary Table 2). Haem can help capture and transport the small amount of oxygen available under HS and it is an essential component of the ETC[63].

Despite the presence of transcriptomic changes explaining HS adaptation and the OS sensitivity cost, there was little overlap between differentially expressed genes and genomic targets of selection, except for *pyk1*, *pzh1* and *ubp9*. For instance, the multi-hit gene *clg1*, which encodes a positive regulator of autophagy[39,72], is not differentially expressed between populations with different OS sensitivity, but 22 other unmutated genes involved in autophagy have changed expression (Supplementary Table 2). Autophagy might be a pathway that contribute to cross-resistance to both HS and OS by recycling biomolecules and removing damaged cell components[73,74]. As for the multi-hit gene *pyk1*, it is downregulated by 36.5% in populations with increased OS sensitivity (Extended Data Fig. 8). Adaptation to HS may require decreasing the glycolytic flux allocated to respiration[32].

The lack of overlap between selection targets at the genomic and transcriptomic levels suggests that the overall changes in expression levels shared by many populations are driven by *trans*- rather than *cis*-regulatory effects. However, interpreting metabolic fluxes from transcriptomics data remains challenging and further exploration into the mechanisms affected by this rewiring is needed to reveal more deeply how they relate to adaptation in our environment.

## Discussion

Experimental evolution is a powerful tool with which to decipher molecular pathways under selective constraints[3,11]. An important result from studies exploiting asexual laboratory evolution is that evolution is usually not limited by mutation: adaptation usually proceeds rapidly and

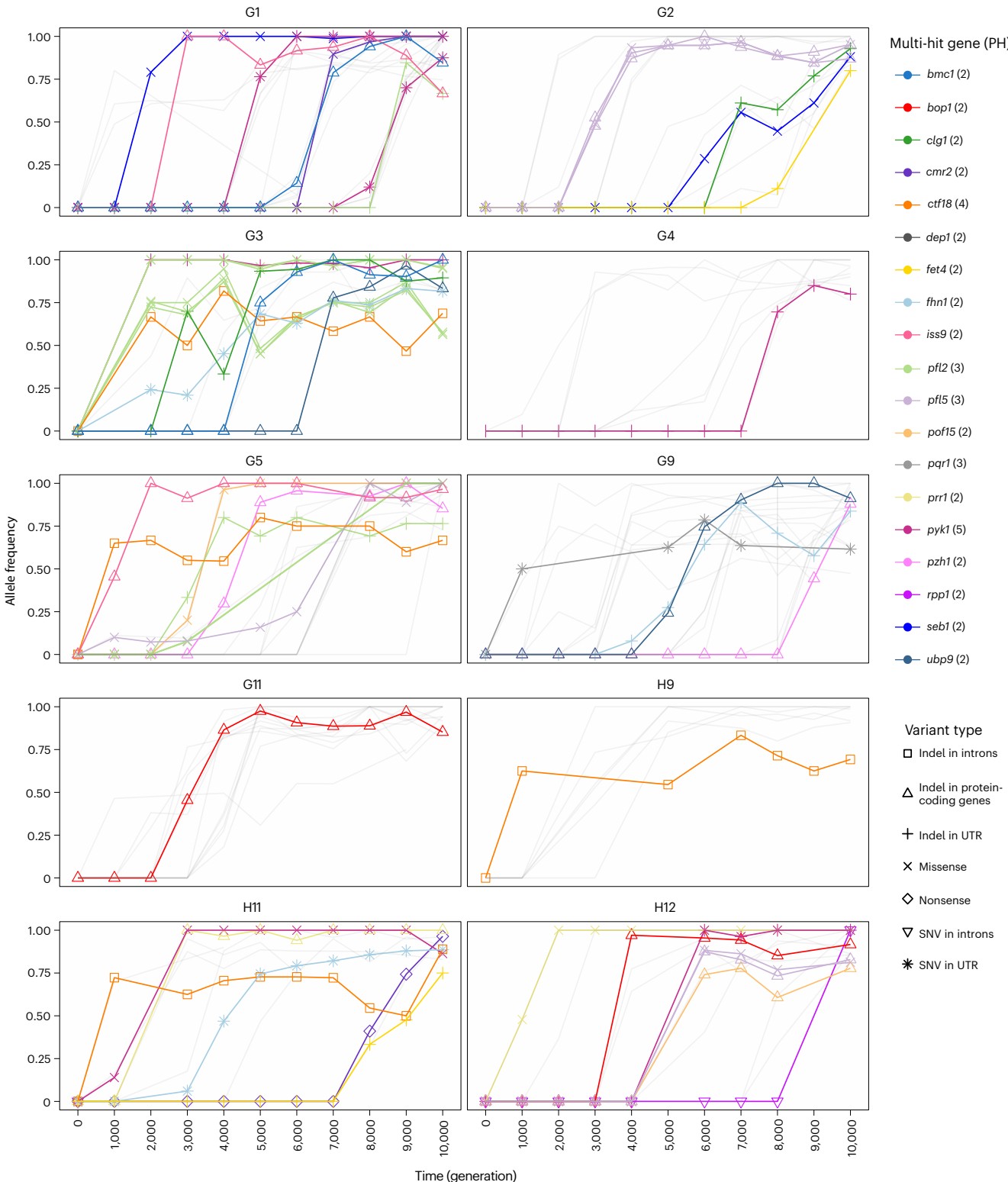

**Fig. 3 | Time series of the allele frequencies of hits in evolved *S. pombe* populations and multi-hit genes.** We defined a 'hit' as a fixed genic variant that is not synonymous (missense, nonsense and indels) and a multi-hit gene as a gene having hits in several populations. In each population, variants that were never fixed or that are not located in multi-hit genes are represented by more transparent lines. PH: number of populations in which the gene is hit.

is often dominated by large-effect mutations. Gene content variation (through ploidy changes) and LOF mutations occur frequently. Perhaps the most interesting result from many of these studies is that laboratory evolution can frequently recapitulate evolutionary outcomes from wild, uncontrolled evolution. Whole-genome duplication, LOF

mutations, the rise of mutators, stable co-existence and, as here, the change of carbon flow towards increased fermentation, are all phenomena thought to have occurred in various species at various times during their evolutionary history, but also notably in cancer cells[11,75–77]. The fact that cancer cells are also evolving populations suggests that

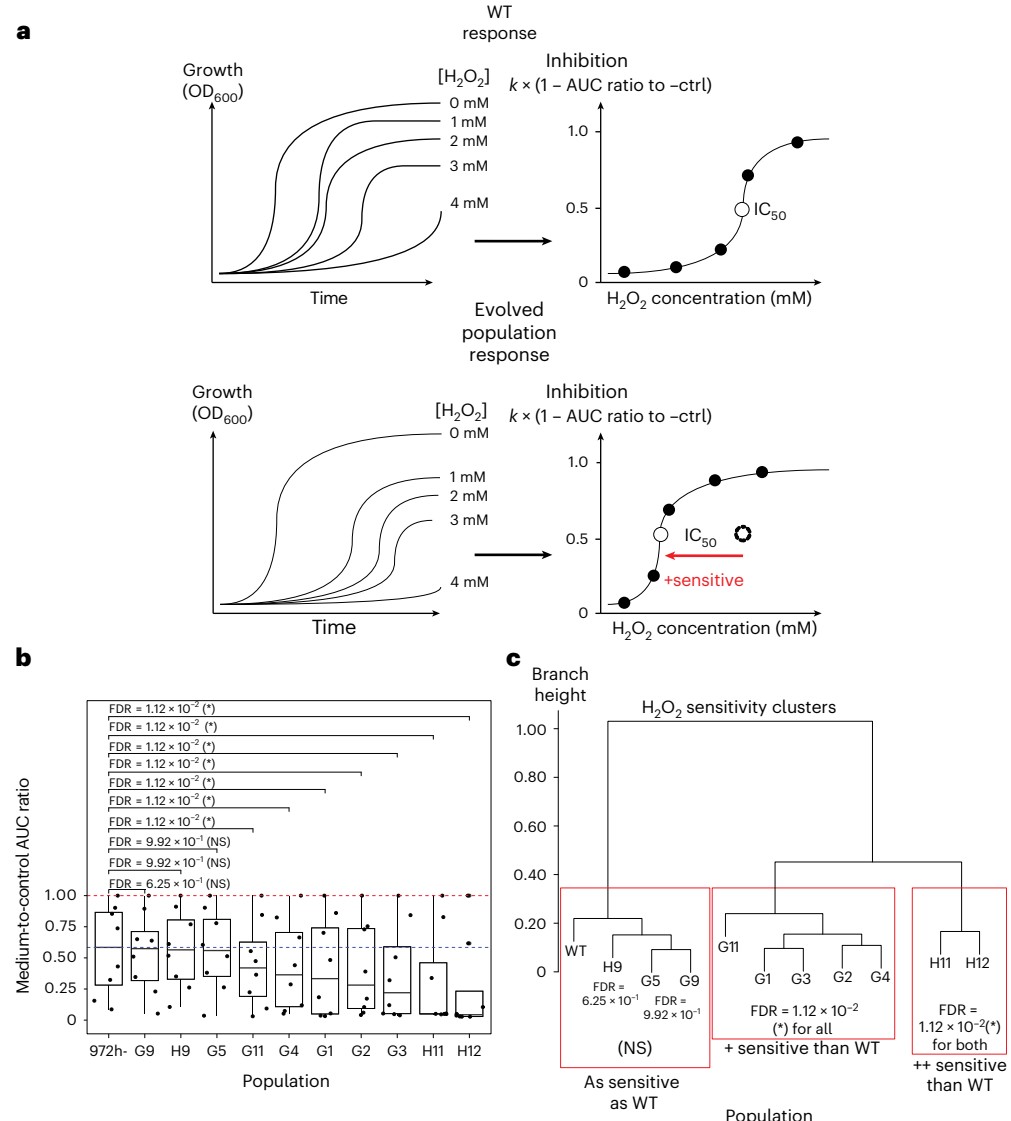

**Fig. 4 | Comparison of the sensitivity levels to $H_2O_2$ between *S. pombe* WT and evolved populations. a**, Schematic illustrating the expected difference in sensitivity between WT and evolved populations. We inferred the degree of inhibition at each concentration of $H_2O_2$ based on the expression in equation (4) and the sensitivity of each population based on dose–response analysis (Methods). The $IC_{50}$ recapitulates the level of sensitivity of a population to $H_2O_2$. Lower $IC_{50}$ values can be interpreted as higher sensitivity to $H_2O_2$, which is what we expected for the evolved populations. **b**, The ratio between the AUC of a population at a certain $H_2O_2$ concentration + YPD and the AUC in YPD without $H_2O_2$. Each point is a different concentration. This is another metric of sensitivity with an expected maximum of 1 (red dotted line). The blue dotted line represents the median sensitivity of the WT. The WT AUC ratio is

compared with each evolved population using a Wilcoxon paired signed rank test. The one-sided adjusted *P* values (false discovery rates (FDR); alternative hypothesis = 'WT AUC ratio is greater than evolved populations') of this test are encoded as follows: \*\*\*from 0 to 0.001 exclusively, \*\*from 0.001 to 0.01 exclusively, \*from 0.01 to 0.05 exclusively, '.' from 0.05 to 0.1 exclusively and non-significant (NS) otherwise. Middle of boxplot represents the median, boundary of boxes represents the 25th and 75th percentile (interquartile range), and whiskers represent the largest or smallest data point that is no farther than 1.5 times the interquartile range from box boundaries. Representative data from one of three biological replicates is shown with the same result. **c**, Red boxes: population clusters based on AUC ratios.

experimental evolution in different single-cell organisms can reveal fundamental principles of the molecular basis of adaptation. The pervasiveness of parallel evolution in the tree of life could be explained by deep conservation of interconnected biological pathways.

Our study extends the discovery of shared molecular mechanisms of adaptation across scales in diverse biological systems. On specific selection targets, mutations that alter iron metabolism, autophagy regulators and vesicle transporters, and that lead to rewiring of the glycolytic flux are common observations in populations adapting under oxygen limitation[78,79]. The orthologs *pykF* and *pyk1* in *E. coli* and *S. pombe*, respectively, are both hit recurrently across several populations, illustrating the conservation of selection targets at the gene level

across distant species. Finally, parallel evolution within model systems is frequent and has been observed in nearly all experimental evolution studies to date, but the precise observed trait under selection varies (examples in refs. 6,80 and reviewed in ref. 81).

Despite common evolutionary outcomes, disparities can arise due to differences in genome architecture and contingency. One of the main findings in evolutionary dynamics is that low-fitness populations typically adapt faster than high-fitness populations. This observation has been seen in several species, and in several environmental contexts[3,6,11]. In the evolution environment of our study, the 'ancestor' *S. cerevisiae* is much fitter than its *S. pombe* counterpart. Yet, the fitness gains by our *S. pombe* populations are relatively modest (Extended Data Fig. 1).

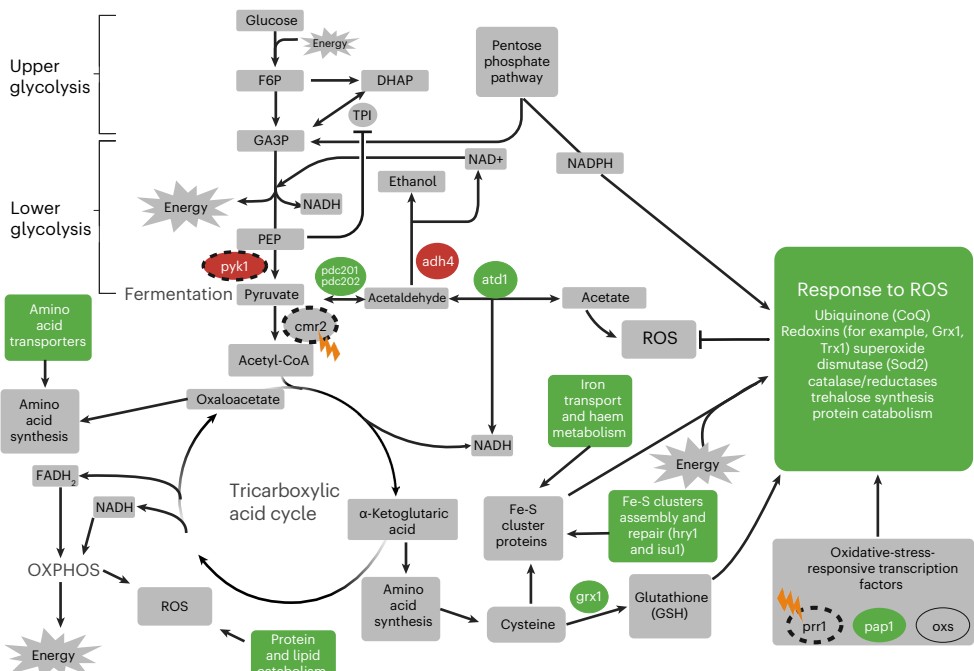

**Fig. 5 | Fission yeast cellular responses to HS in populations with increased OS sensitivity.** Transcriptomic changes are determined by comparing the populations with increased OS sensitivity with populations with similar OS sensitivity to that of WT whereas genomic changes are obtained by comparing these evolved populations with WT (Methods). Green fill: upregulation, red fill: downregulation. When processes are colour filled, they represent cases where some genes involved in those processes were upregulated. Grey fill: molecules or processes that were not associated with a gene with significant effect or not measured (no data). Orange lightning symbol: LOF hits in H11—the only population with such variants in several multi-hit genes (dotted circles). Figure created in BioRender; Arnaud, N. https://biorender.com/8pi06vc (2026).

We propose that evolutionary contingency may explain this 'evolvability'. Indeed, it was found previously that the identity of specific molecular pathways of gene deletions had a strong impact on fitness gain after evolution[13].

Further supporting this, the most frequent targets of adaptation in the previous *S. cerevisiae* study were genes involved in adenine biosynthesis and the mating pathway, for which genomic hits have high-fitness effects[3]. However, these pathways are not selected in these *S. pombe* populations. Based on our understanding of the molecular pathways in both species, mutations in adenine biosynthesis genes were not expected in our ade+ *S. pombe* populations. In contrast, the *S. cerevisiae* populations were *ade2⁻* and accumulate a toxic intermediate that can be suppressed by other adenine pathway mutations. Similar effects can be observed in *ade6⁻* strains of *S. pombe*[82]. Similarly, mutations in the mating pathway are not expected in *S. pombe* due to its preferential haploid lifestyle and its inactivity in rich media[23,24].

Another pathway mutated frequently in the budding yeast laboratory evolution experiment is the Ras-cAMP pathway[3,8,83]. Observed LOF mutations in *IRA1* or *IRA2* (but usually not both), relieve inhibition of Ras and lead to PKA activation[3,84]. Activation of PKA ultimately increases glycolysis through phosphorylation of Pyk1 ref. 85). Our *S. pombe* populations never mutated the *IRA1/2* orthologue (*gap1*), although it is a single-copy gene in fission yeast. More surprising are the *pyk1* mutations; most were found in the regulatory region of the gene, which we believed initially to be an analogous way for *S. pombe* to upregulate Pyk1 activity. However, whole-transcriptome analyses suggest that these mutations downregulate Pyk1 activity (Fig. 5). In our dataset, three populations in which *pyk1* is downregulated have *cis*-regulatory mutations around the gene, whereas G2 is the only population in which *pyk1* is downregulated without such mutations (Extended Data Figs. 7 and 8). Mutations leading to lower Pyk1 activity have been observed in wild *S. pombe* populations and in the common laboratory strain[35], suggesting that decreasing Pyk1 activity may be adaptive. In ref. 35,

the authors showed that a decrease in Pyk1 activity does not increase growth rate or biomass, contradicting the adaptive signature from parallel evolution. However, it was noted that lower Pyk1 activity might be an adaptive trade-off as lower glycolytic flux increases OS resistance through higher pentose-phosphate pathway flux and NADPH production. However, our *S. pombe* populations were often more sensitive to OS. This is explained potentially by the fact that the slightly reduced glycolytic flux is redirected mostly toward fermentation, which is upregulated to compensate for the fission yeast respiration deficiency in hypoxic conditions.

Adaptation to our growth environment has led to an increased OS sensitivity for most evolved populations through several molecular mechanisms. First, changes in Pyk1 activity and LOF mutations in enzymes responsible for the transition between glycolysis and oxidative phosphorylation lead to increased intracellular ROS through higher ethanol production and a deficiency in the production of energy, which sustains the redox metabolism[34,44]. Reducing Pyk1 activity is also consistent with the other mutations that modulate central carbon metabolism (Figs. 3 and 5). Second, *S. pombe* cells are sensitive to polypeptones, a principal component of our environment[59]. Mutations that lower flux through the mitochondrial ETC increase resistance to polypeptones because this sensitivity requires CoQ10 (ref. 59). Strikingly, there is also evidence in *S. japonicus* that lower glycolysis is rewired frequently in natural populations[32]. Thus, adaptation to HS or polypeptones in petite-negative yeast species with modest fermentation rates might often come at the expense of costly carbon flow rewiring, hindering the response to OS.

Several genomic variants and transcriptomic changes were involved in the apparent trade-off between HS and OS. Fermentation upregulation and LOF in lower glycolysis and genes involved in the response to OS may explain the increased sensitivity to OS. However, some genes involved in OS response, autophagy and vesicle transport may contribute to cross-resistance to both stresses[86–90] and indeed we

observed three populations (G5, G9 and H9) with hits in those pathways that did not seem to have increased OS sensitivity, and those populations retain WT-like expression level of the lower glycolysis pathway. This cross-resistance is also observed in other species like *Kluyveromyces lactis, S. cerevisiae* and human cancer[91–94]. The fact that this trade-off has not been observed in budding yeast may be explained by its historical exposure to several anaerobic environments and better adaptations to HS[34]. In contrast, *S. pombe* and *S. japonicus*, for which lower glycolysis is less active, tend to adapt to HS through LOF in lower glycolysis and upregulation of fermentation and upper glycolysis[32,35]. Furthermore, the fact that the response to HS involved mostly transcriptomic changes in genes that were not mutated suggests that the adaptive transcriptomic changes observed across populations were driven by *trans*-regulatory rather than *cis*-regulatory effects. This also shows that the transcriptome can exhibit evolutionary parallelism.

This study shows that several mechanisms are involved in adaptation and that contingency can lead to diverse evolutionary outcomes, even within a single species and under the same conditions. One way that future research could better predict adaptive trajectories under stress conditions is by leveraging the presence of rapid transcriptomic changes in response to a new stress independently of mutations. These responses could provide hints toward adaptive cellular changes that could later be cemented by mutations with similar effects[11,95].

## Methods

### Experimental evolution and sequencing
We evolved 15 *S. pombe* populations of the laboratory strain 972h- at 30 °C in wells of a lidded, flat-bottom 96-well plate containing 128 µl of YPD plus antibiotics (1% yeast extract, 2% peptone, 2% dextrose, 100 µg ml$^{-1}$ ampicillin, 25 µg ml$^{-1}$ tetracycline) for 10,000 generations, collecting and freezing samples in 27% glycerol at −80 °C every 70 generations. The remaining wells contained blank controls or populations of other yeast species and will be described elsewhere. This is the same procedure used to generate the compared *S. cerevisiae* dataset[3]. Because these conditions yield a doubling rate of ten generations per day, we performed a 1 in 1,024 serial dilution every day (1/2$^{10}$). This corresponds to an effective population size of ~$6.00 \times 10^4$ and a bottleneck population size of ~$8.00 \times 10^3$ (refs. 3,96). We chose to perform whole-population whole-genome sequencing every 1,000 generations, and we were able to grow back cells from 93 out of these 100 samples after storage and sequenced these time points with Illumina NovaSeq paired-end sequencing (mean effective coverage ~40×). Sequencing and DNA preparation was performed as in the *S. cerevisiae* dataset[3].

### Measuring fitness in the *S. pombe* populations
We grew the evolved populations, the WT and a *S. cerevisiae* reference strain expressing green fluorescent protein (YAN438: *MATα his3Δ1 ura3Δ0 leu2Δ0 lys2Δ0 can1::RPL39pr-ymGFP_STE2pr-SpHIS5_STE3pr-LEU2*) for 24 h. In contrast to our previous study[3], this *S. cerevisiae* strain does not express a killer phenotype as it does not possess ScV-M1 killer-virus RNA. Next, because the green *S. cerevisiae* strain grew much faster than *S. pombe*, we prepared mixtures of the 'dark' *S. pombe* populations and the green budding yeast at a 9:1 ratio and grew the mixed cultures for 24 h. This ratio also reduces potential species interactions between *S. cerevisiae* and *S. pombe*. We then propagated each mixture serially for 3 days and used a Beckman–Coulter CytoFlex flow cytometer to measure the frequency of the 'dark' *S. pombe* and the green *S. cerevisiae* strain daily. All particle counts were acquired at 30 µl s$^{-1}$ rate for a maximum of 60 s or until 10,000 events were detected from a 96-well plate containing the mixtures. The data were analysed using the CytoFlex acquisition and analysis software CytExpert (v.2.5). The 'dark' *S. pombe* cells were distinguished from the green *S. cerevisiae* strain using FITC-A/PE-A gates. This process was replicated on a different day for biological replicates. Using the cell counts, we then calculated the frequency of each population in the mixture and inferred fitness

based on the slope of a linear regression between the logarithm of the frequency ratio and the number of generations:

$$\ln \left( \frac{\text{frequency}_{S.pombe}}{\text{frequency}_{\text{green }S.cerevisiae}} \right) = (\text{Fitness} \times \text{number of generations}) + \text{intercept} \quad (1)$$

The final fitness was the average of the two biological replicates.

### Variant calling
We used Varscan (v.2.4.6)[97] to call single-nucleotide variants and indels. Because contamination, sequencing errors and low-complexity regions can affect the variant calling, we applied some filters to the called variants (Supplementary Table 3). These filters ensure that the calls are supported by reads, are time-consistent and are not affected by possible errors in the reference genome, for example, avoiding confusion between ancestral alleles that were not present in the reference genome (972h-) and fixed de novo mutations. Briefly, the filters used mapping and read base quality, allele frequency and absence of strand biases. In addition, because of the relatively sparse sequencing of our experiment (every 1,000 generations), most genuine mutations will be fixed in cohorts or throughout the experiment, we also used a filter for genetic hitchhiking and clonality.

When applying these filters, we also found 11 cases where the same recurrent indel in low-complexity region was called in several populations. Although these can be signatures of strong mutation bias or strong selection, they can also indicate evidence of cross-contamination in the evolution or DNA sequencing, as well as mis-indexed reads or read mapping issues. To rule these out, we applied several manual inspections. First, when mutations arise in a population and fix, they must remain fixed for the rest of the evolution. Second, if a recurrent mutation fixes in two populations, they must not share other mutations substantially within the samples and the rest of the time course (which would indicate cross-contamination). Third, other populations must not show strong evidence of these mutations (which would indicate some sequencing bias). Although this does not rule out all possible issues with long-term evolution experiment and robotic passaging, or other artefacts of our variant-calling pipeline, these mutations were still considered in our analyses as they may reveal mutational biases or other interesting phenomena (such as genetic-level nucleotide convergence). Despite these filters, three variants remain ambiguous (in *dep1*, *rpp1* and *pqr1*). A version of Fig. 3 without these recurrent mutations is found in Extended Data Fig. 5.

We also found cases of mutations that seemed to remain at intermediate frequencies, one of which was recurrent (for example, *ctf18*). This could indicate some form of frequency dependence or aneuploidy, but it can also be a signature of a read mapping artefact. To explore this, we inspected the mapped reads manually and found that these occurred in large polynucleotide tracts. As reads may not always span the whole tract, variant-calling algorithms may fail to capture indels as fixed. We decided to leave the estimated frequencies as is.

Finally, complex mutations, such as events that may replace several adjacent codons simultaneously, may be due to alternative read mapping locations or due to complex short-tract recombination. We used the mapping quality filter to remove cases of alternative read mapping locations, and thus these complex mutations were left as is by the variant-calling pipeline as they may be genuinely multiple mutations that get fixed in the same region. Nevertheless, this means that some categories of mutations may be inflated due to incorporation from a complex event.

### Identifying signatures of selection from the substitution spectrum
We defined substitutions as variants with an allele frequency ≥75%. Then, we compared the substitution spectrum of each population with

the one expected after 10,000 generations of evolution under a neutral null model. The number of fixed variants in each mutation category is calculated based on the species' evolution rate, genome size, genome architecture (for example, ratio of coding to non-coding regions in the genome) and the proportion of single-nucleotide mutations derived from the genetic code and associated with a specific category of mutations. This allowed us to derive two equations:

$$\text{No. of subs in category } X_{nm} = S_{rate} \times t \times G_{size} \times r_x \times p_x \tag{2}$$

$$\text{No. of indels in category } X_{nm} = I_{rate} \times t \times G_{size} \times r_x \tag{3}$$

where No. of subs in category $X_{nm}$ is the expected number of substitutions in a mutation category $X$ assuming a neutral model of evolution, No. of indels in category $X_{nm}$ is the expected number of indels of type $X$ assuming a neutral model of evolution, $S_{rate}$ is the species substitution rate, $I_{rate}$ is the species indel rate, $t$ is the number of generations observed since the start of the experiment, $G_{size}$ is the species genome size, $r_x$ is the proportion of the genome covered by protein-coding or non-coding sites, depending on the mutation category $X$ and $p_x$ is the proportion of single-nucleotide mutations derived from the genetic code that can be classified as mutations from category $X$.

Next we sought signatures of parallelism and positive selection in protein-coding genes among the fixed variants. We defined a 'hit' as a fixed variant that is not synonymous (missense, nonsense or indel), and a multi-hit gene as a gene with hits in several populations (in at least two out of ten populations). We then compared the number of population hits (PH) in the set of multi-hit genes with the PH expected under a null model, assuming that all genes evolve and fix variants at the same rate. The expected PH was determined based on a number drawn from a Poisson distribution where the argument $\lambda$ (average number of events per interval) is the product of the substitution rate of *S. pombe*, the gene length, the generation time, the number of populations and the proportion of fixed variants that are not synonymous as determined in equations (2) and (3).

### Evaluating and comparing sensitivity to $H_2O_2$

We grew the ten evolved *S. pombe* populations and the WT ancestor in the rows of a 96-well plate at 30 °C for 36 h. The medium was YPD plus $H_2O_2$ concentrations of 0 mM, 0.5 mM, 1 mM, 1.5 mM, 2 mM, 2.5 mM, 3 mM and 3.5 mM. Each row of the 96-well plate corresponded to a different concentration of $H_2O_2$. The growth of each population was estimated using the optical density $(OD)_{600}$ time series measured from a BioTek Synergy HTX plate reader. For each time series, the area under the curve (AUC) was measured using the R package package growthcurver (v.0.3.1)[98]. Using the AUC as a proxy for growth, we then compared the AUC of each population at different $H_2O_2$ concentrations with the AUC obtained when $H_2O_2$ is absent from the medium. Next we inferred the degree of inhibition with:

$$\text{Inhibition} = k \times \frac{AUC_{medium}}{AUC_{control}} \tag{4}$$

where $k$ is a normalizing factor that makes sure the inhibition metric maximum is 1, AUC is the area under the growth curve, $AUC_{medium}$ represents the growth at a certain concentration of $H_2O_2$ and $AUC_{control}$ represents the growth in a YPD medium without $H_2O_2$. This expression is correlated negatively to the AUC ratio and ranges from 0 to 1. Using the dose–response analysis R package drc (v.3.0-1)[99], we then fitted a log-logistic model to this metric and $H_2O_2$ concentrations to estimate the half maximal inhibitory concentration $(IC_{50})$, which recapitulates the level of sensitivity of a population to $H_2O_2$. Data in the manuscript are representative results of the less noisy biological replicate as determined by curve fit. Furthermore, we found that the $H_2O_2$ concentration tended to vary due to instability of the chemical,

which would at times shift the $IC_{50}$, without changing the interpretation of the experiment. Finally, we performed a cluster analysis on the populations based on their distribution of AUC medium-to-control ratio at different concentrations. We used the hclust function from the package stats v.4.1.2 (ref. 100) to perform the clustering and the composition of each cluster was identical no matter the distance metric tested (Gower, Manhattan, Euclidean, Bray-Curtis), which we obtained with the function vegdist from the package vegan (v.2.7-2)[101]. Growth curve figures are also shown in Extended Data Fig. 6.

### RNA extraction, sequencing and transcriptomic analysis

We extracted RNA of the evolved populations in two biological replicates using an in-house protocol that works well with *S. cerevisiae*. Briefly, cells were grown to mid-log phase in YPD by diluting overnight saturated cultures and growing for 4–6 h before lysing using 100 µl of 5 mg ml⁻¹ Zymolyase + 10 mM dithiothreitol, incubated at 37 °C for 5 min. The weakened cells were then incubated with 100 µl of 2% SDS for a final SDS concentration of 1%. Lysis buffer was added (200 µl; 4.5 M guanidine thiocyanate, 10 mM EDTA) and mixed gently by inverting. Lysate clarification indicated complete lysis. The SDS was then precipitated using 200 µl of 3 M potassium acetate at pH 5.

After centrifugation, the lysate was passed through a minipreparation silica column to bind DNA. Under these conditions, RNA does not bind to the silica columns. Therefore, we collected the flowthrough. We then mixed it with an equal volume of pure isopropanol and passed it through another minipreparation silica column (which will now bind RNA). The column was then washed with 400 µl of wash buffer (10% guanidine thiocyanate, 25% isopropanol and 10 mM EDTA) and then twice with 600 µl of 80% ethanol. After drying the column, the RNA was eluted with 50 µl of 10 mM Tris pH 8.5. RNA integrity was verified on an agarose gel. One population (H12) failed to get sufficiently high-quality RNA for sequencing.

We sequenced the successfully extracted RNA using NEBNext Ultra II (catalogue no. E7770) on an Illumina NovaSeq X paired-end sequencing lane. This yielded $10.6 \times 10^6$ 150-bp paired-end reads per sample on average $(\sigma = 1.32 \times 10^6)$. We pre-processed the reads with fastp (v.0.23.1) with the arguments --trim_poly_x --trim_poly_g --cut_front --cut_front_mean_quality 20 --cut_tail --cut_tail_mean_ quality 20 --detect_adapter_for_pe[102]. Next we mapped the reads to the reference (972h-) using Samtools (v.1.17), removed duplicates with Picard (v.2.26.3) and counted the number of reads per gene using featureCounts from sub-read (v.2.0.6)[103–105]; counts were converted to fragments per kilobase of transcript per million mapped reads. We then performed principal component (PC) analysis of the transcriptome and found two main clusters (PC1 and PC2 explained about 75% of the transcriptomic variance), one which contained the WT and the other contained only populations that were more sensitive to OS than WT. One population that was as sensitive as WT clustered independently, and two populations that clustered with the WT were more sensitive to OS. Upon analysis of their gene hits, we found that they have LOF mutation in genes that relate to OS resistance.

To determine which genes and their expression differences may be responsible for the increased OS sensitivity of most populations, we used the DESeq2 package (v.1.34.0) to detect differentially expressed genes between the two clusters (that is, populations that are as sensitive and more sensitive than WT)[106]. Gene functional categories and gene ontology annotations were obtained from Pombase data[39].

### Reporting summary

Further information on research design is available in the Nature Portfolio Reporting Summary linked to this article at https://doi. org/10.1038/s41559-026-03017-1.

## Data availability

Raw DNA and RNA-sequencing data are available on NCBI, Bioproject database with accession code PRJNA1315931. Source data are provided with this paper.

## Code availability

Analysis code for this project is available via Zenodo at https://doi.org/10.5281/ZENODO.18510578 (ref. 107).

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

## Acknowledgements

We thank all members of the M.M.D. and A.N.N.B. laboratories for helpful comments on the manuscript. We thank the Northwest building staff, the Bauer Core staff, the CAGEF and TCAG sequencing cores for technical support during the work. A.N.'G. acknowledges funding from Canada Graduate Research Scholarship from NSERC. A.N.N.B. acknowledges funding from NSERC (RGPIN-2021-02716), CIHR (OGB-185738), Connaught and the OVPRI at UTM. M.S.J., E.R.J., K.K. and K.R.L. acknowledge the National Science Foundation for Graduate Fellowship funding. K.R.L. acknowledges the Simons Foundation (DMS-1764269) and the Hertz foundation. J.G. acknowledges PRISE from Harvard University. C.W.B. acknowledges the National Defense Science and Engineering Graduate fellowship. R.P. acknowledges the Department of Biotechnology, Ministry of Science and Technology for Boston Bangalore Biosciences Beginnings programme. M.J.M. acknowledges funding from the Australian Research Council (FT170100441). M.M.D. acknowledges funding from the Simons Foundation (376196), NSF (PHY-1914916) and NIH (R01 GM104239).

## Author contributions

Conceptualization: A.N.'G., V.W., C.W.B., J.B., G.Y.-S.B., M.E.D., S.G., J.G., M.G., C.M.H., P.T.H., T.J., E.R.J., M.S.J., K.K., K.R.L., J.M., A.M., S.V.P., A.M.P., J.C.P., R.P., A.R.-C., T.R.-B., C.T., M.J.M., M.M.D. and A.N.N.B. Investigation: A.N.'G., V.W., C.W.B., J.B., G.Y.-S.B., M.E.D., S.G., J.G., M.G., C.M.H., P.T.H., T.J., E.R.J., M.S.J., K.K., K.R.L., J.M., A.M., S.V.P., A.M.P., J.C.P., R.P., A.R.-C., T.R.-B., C.T., M.J.M., M.M.D. and A.N.N.B. Formal analysis: A.N.'G. and A.N.N.B. Writing—review & editing: A.N.'G., J.B., S.V.P., T.R.-B., C.M.H., M.M.D. and A.N.N.B.

## Competing interests

The authors declare no competing interests.

## Additional information

**Extended data** is available for this paper at https://doi.org/10.1038/s41559-026-03017-1.

**Correspondence and requests for materials** should be addressed to Alex N. Nguyen Ba.

Arnaud N'Guessan [1], Vivian Wang [1], Christopher W. Bakerlee[2,3,4], Jenya Belousova [2], Greta Y.-S. Brenna [2], Megan E. Dillingham[5], Shreyas Gopalakrishnan[2], Juhee Goyal[2,6], Misha Gupta[2], Caroline M. Holmes[2], Parris T. Humphrey[2,4], Tanush Jagdish[2,4,5], Elizabeth R. Jerison [2,7,8], Milo S. Johnson [2,4], Katya Kosheleva[2,8], Katherine R. Lawrence[2,4,9], Jiseon Min[2,3,4,6], Alief Moulana[2], Shreyas V. Pai [2,5], Angela M. Phillips [2], Julia C. Piper[8,10], Ramya Purkanti [2,11], Artur Rego-Costa [2], Tatiana Ruiz-Bedoya [2], Cecilia Trivellin [2], Michael J. McDonald [2,12], Michael M. Desai [2,4,8] & Alex N. Nguyen Ba [1,13] ✉

[1]Department of Cell and Systems Biology, Ramsay Wright Laboratories, University of Toronto, Toronto, Ontario, Canada. [2]Department of Organismic and Evolutionary Biology, Harvard University, Cambridge, MA, USA. [3]Department of Molecular and Cellular Biology, Harvard University, Cambridge, MA, USA. [4]NSF-Simons Center for Mathematical and Statistical Analysis of Biology, Harvard University, Cambridge, MA, USA. [5]Graduate Program in Systems, Synthetic and Quantitative Biology, Harvard University, Cambridge, MA, USA. [6]John A. Paulson School of Engineering and Applied Sciences, Harvard University, Cambridge, MA, USA. [7]Department of Applied Physics, Stanford University, Stanford, CA, USA. [8]Department of Physics, Harvard University, Cambridge, MA, USA. [9]Department of Physics, Massachusetts Institute of Technology, Cambridge, MA, USA. [10]Clover Food Labs, Cambridge, MA, USA. [11]Cancer Research UK, Manchester Institute, Manchester, UK. [12]School of Biological Sciences, Monash University, Clayton, Victoria, Australia. [13]Department of Biology, University of Toronto at Mississauga, Mississauga, Ontario, Canada. ✉e-mail: alex.nguyenba@utoronto.ca

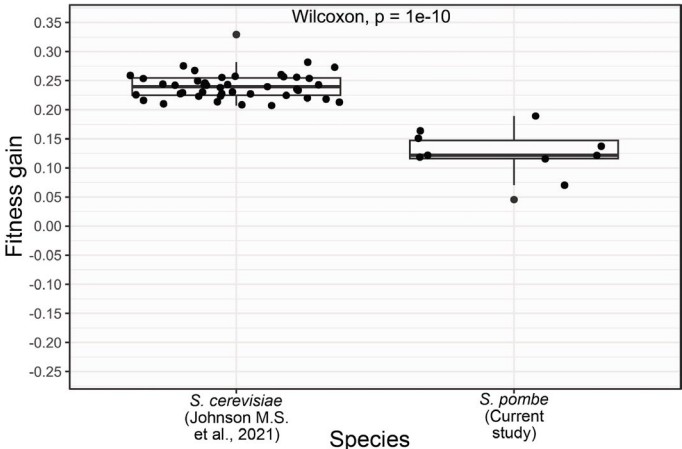

**Extended Data Fig. 1 | Fitness gain measured during experimental evolution of _S. cerevisiae_ and _S. pombe._** The fitness gain is the difference between the fitness at the final time point (10,000 generations) and the ancestral fitness (time 0). The two-sided p-value is obtained from performing a Wilcoxon rank sum test (Mann-Whitney U test). N = 43 independent _S. cerevisiae_ and 10 independent _S. pombe_ populations, each point is the mean of two biological replicates. Middle of boxplot represents the median, boundary of boxes represents the 25th and 75th percentile (interquartile range), and whiskers represent the largest or smallest data point that is no farther than 1.5 times the interquartile range from box boundaries.

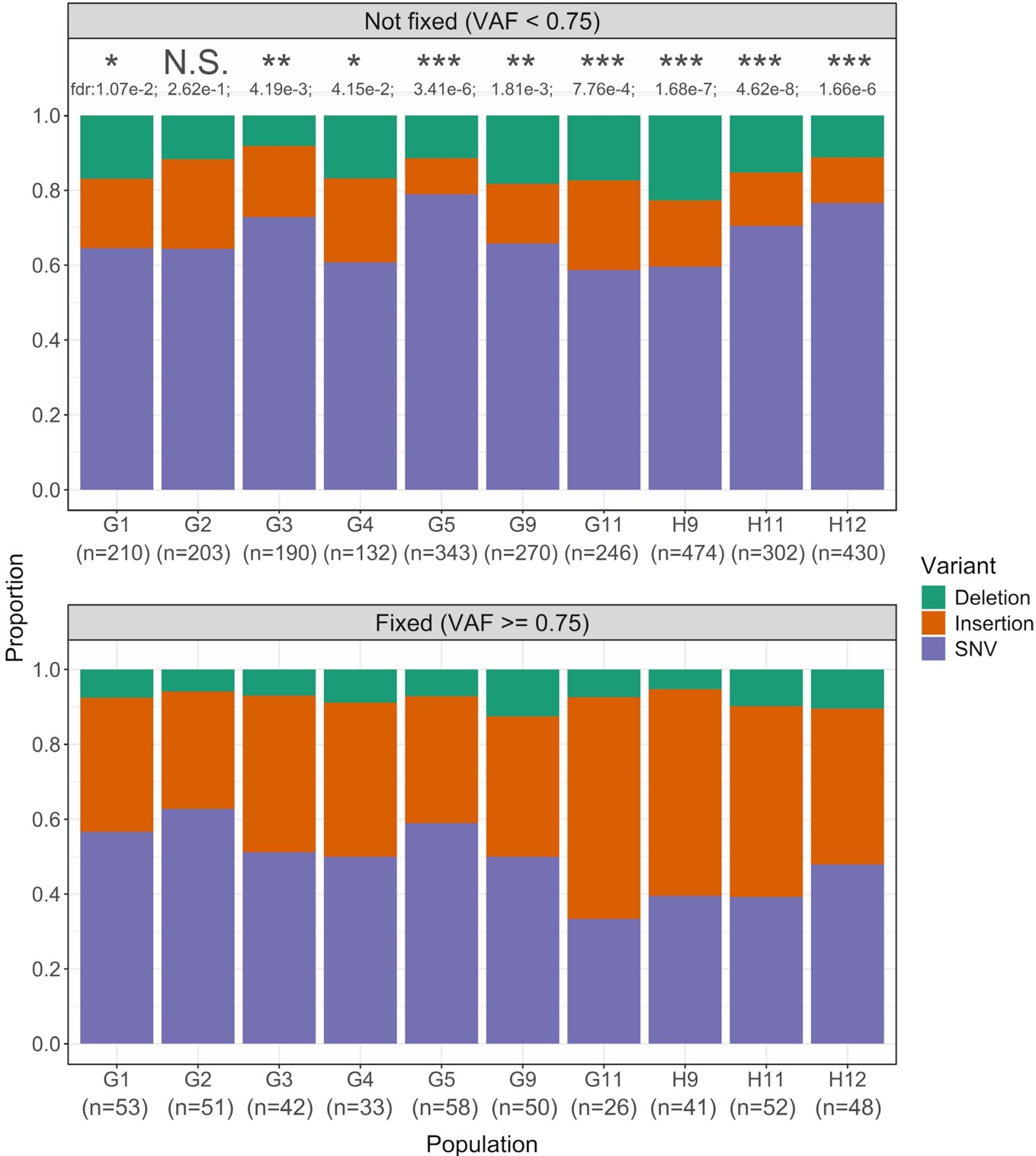

**Extended Data Fig. 2 | *S. pombe* fixation bias.** The fixation bias is defined as the change of frequencies between the fixed and non-fixed variants. The two-sided p-values of a Chi-Squared test comparing non-fixed and fixed proportions are encoded as follows: *** from 0 to 0.001 exclusively, ** from 0.001 to 0.01 exclusively, * from 0.01 to 0.05 exclusively, "." from 0.05 to 0.1 exclusively and N.S. otherwise, which is the acronym of "non-significant".

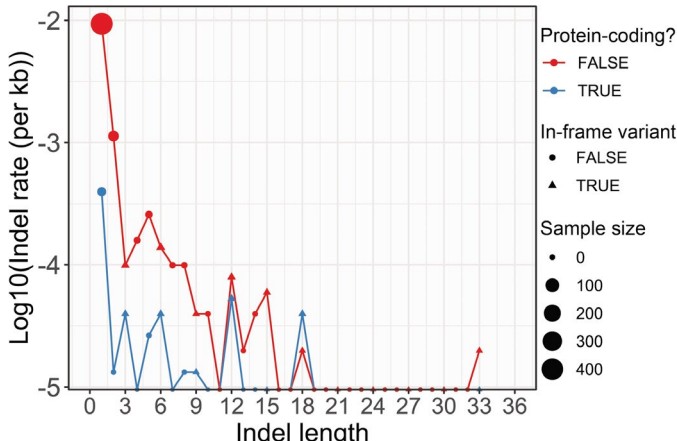

**Extended Data Fig. 3 | Indel rate for coding vs non-coding indels and in-frame vs frameshift indels.** We defined the indel rate as the number of indels per kilobase (Kb) and plotted it on a log10 scale on the y-axis. Indel lengths are in base pair. The dot size represents the size of each indel set.

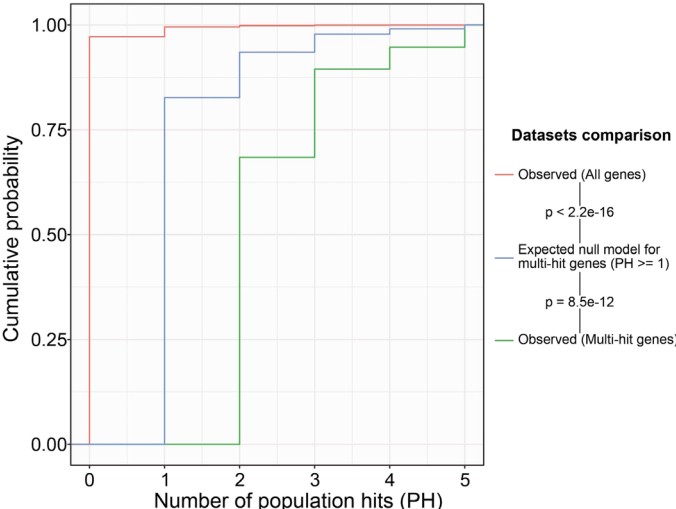

**Extended Data Fig. 4 | The expected vs observed cumulative distribution of population hits across the 10 *S. pombe* populations.** We defined population hits (PH) as the number of populations in which a gene has a hit. The null model assumes that all genes evolve and fix variants at the same rate, so the expected PH is determined by this assumption and follows a Poisson distribution (**Methods**). The one-sided p-values were obtained using a Kolmogorov-Smirnov test (alternative: cumulative distribution function 1 is "higher" than function 2, which means that there are higher values in function 2).

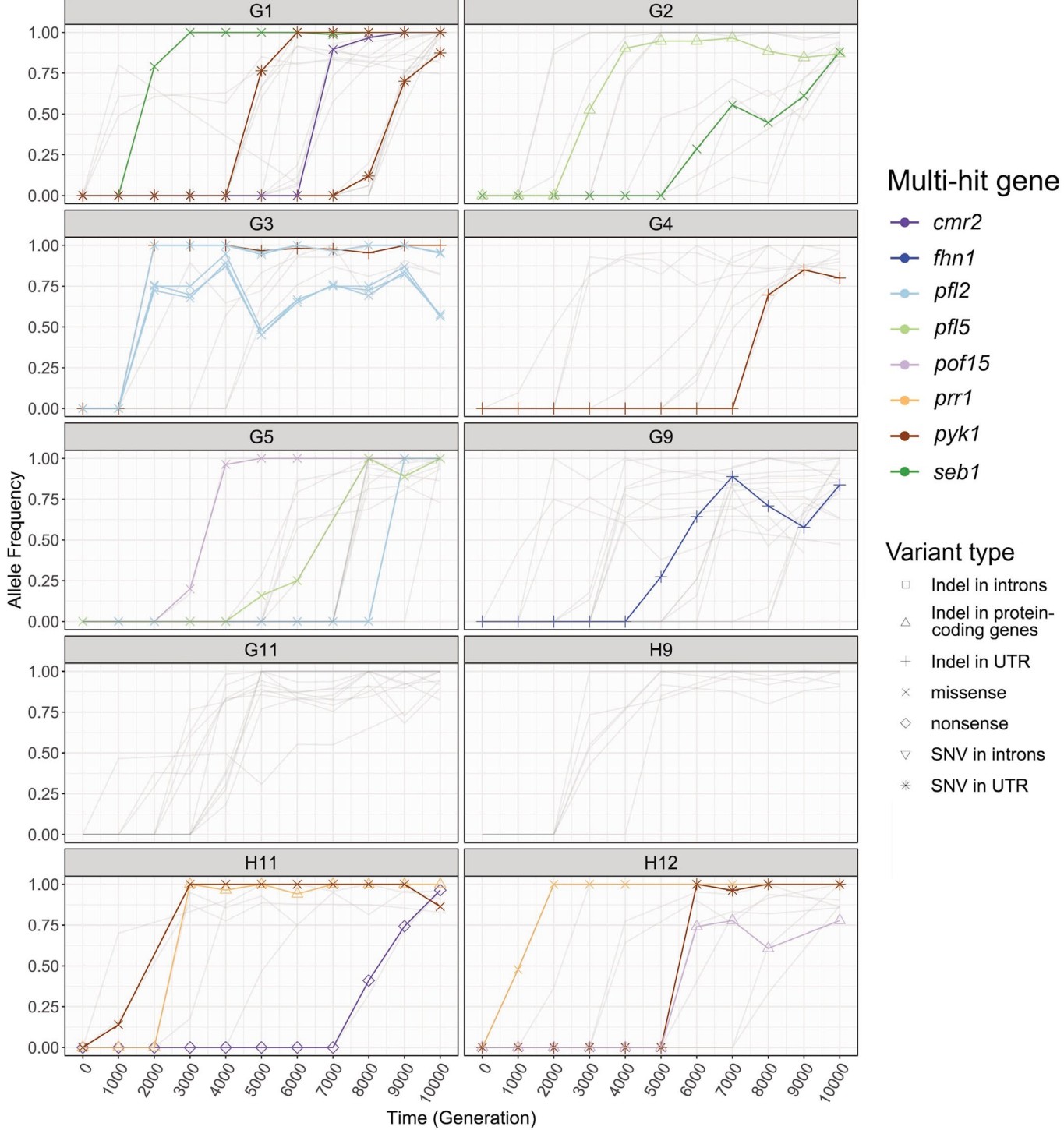

**Extended Data Fig. 5 | Time series of the hits' allele frequencies in evolved S. pombe populations and multi-hit genes without recurrent mutations.** Time series of the hits' allele frequencies in the *S. pombe* populations without recurrent mutations. We defined a "hit" as a fixed genic variant that is not synonymous (missense, nonsense and indels) and a multi-hit gene as a gene having hits in multiple populations. In each population, variants that were never fixed or that are not located in multi-hit genes are represented by more transparent lines. In the legend, PH represents the number of populations in which the gene is hit.

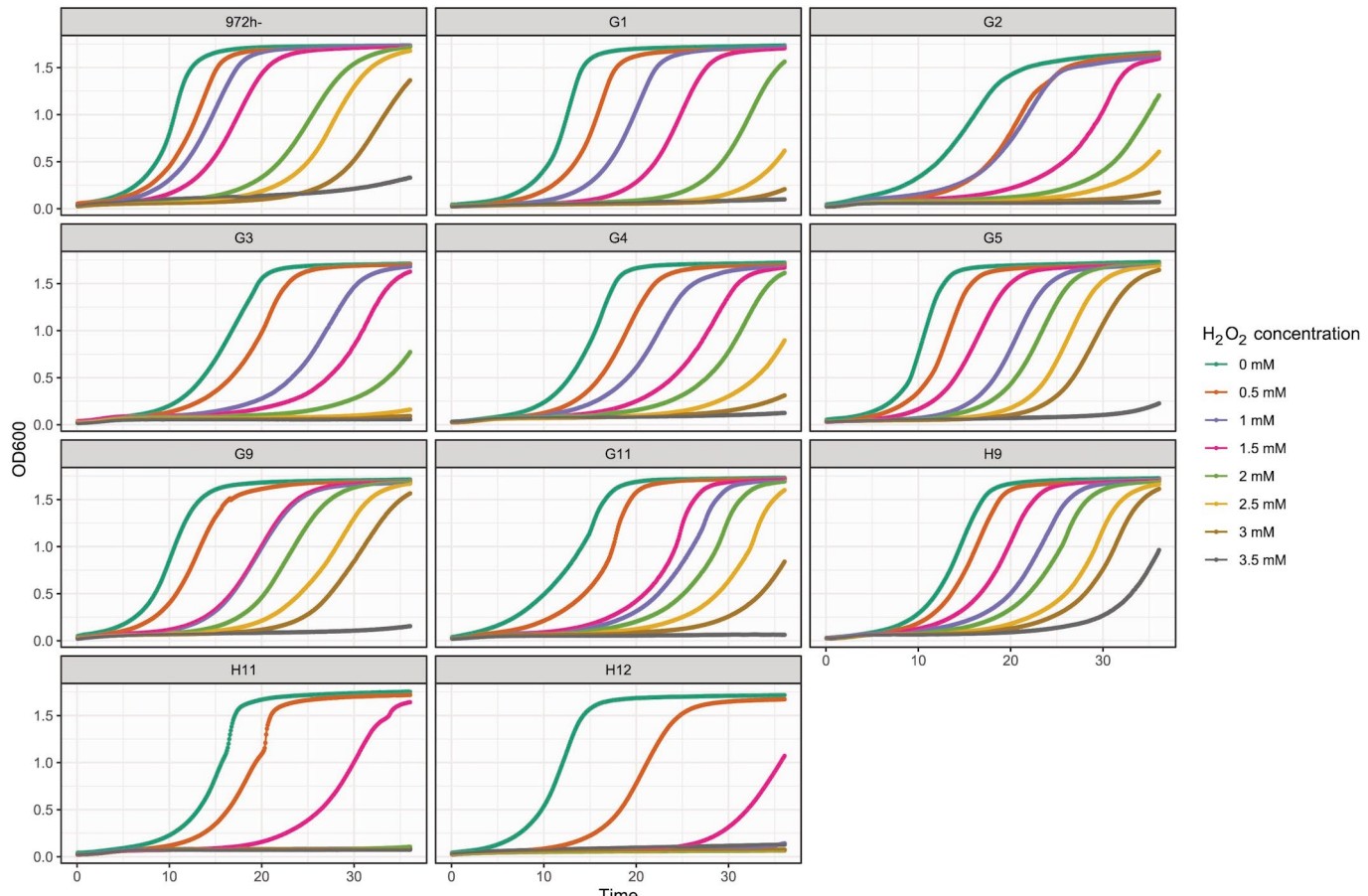

**Extended Data Fig. 6 | Growth curves of evolved populations at different hydrogen peroxide concentrations.** *S. pombe* evolved populations and ancestor were grown at different hydrogen peroxide concentrations for 36 h to determine their resistance to oxidative stress. Growth curves shown are representative data from one or three replicates with the same results.

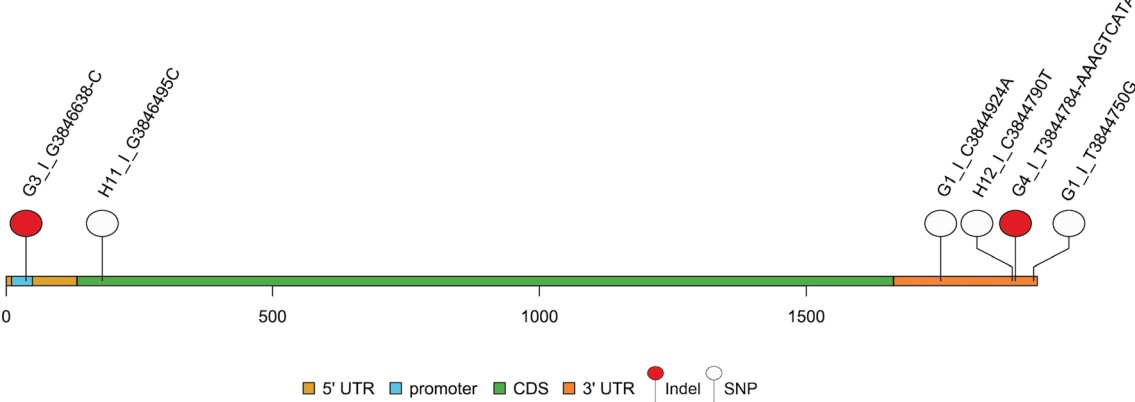

**Extended Data Fig. 7 | *pyk1* mutations across the 10 evolved *S. pombe* populations.** The positions in the pyruvate kinase gene *pyk1* are illustrated on the x-axis (coordinates in base pair) while the coding and non-coding regions have distinct colors. The mutations are red when they represent indels and white when they represent single-nucleotide polymorphisms. The format of the variant name is "population_chromosome_reference allele_position in the chromosome_new allele". For insertions, the alternative allele starts with the character "+" while an alternative allele starting with the character "-" represents a deletion.

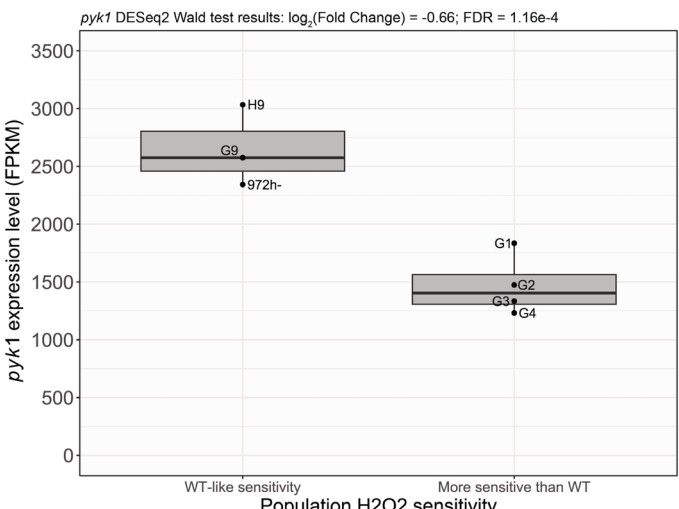

_pyk1_ DESeq2 Wald test results: log$_2$(Fold Change) = -0.66; FDR = 1.16e-4

**Extended Data Fig. 8 | pyk1 expression level in WT-like populations compared to populations that are more sensitive to OS.** Gene expression levels are obtained from read mapping (Methods) using the number of Fragments Per Kilobase of transcript per Million mapped reads (FPKM) metric. We defined WT-like populations as evolved populations with transcriptomic profiles and OS sensitivities similar to WT. Populations that are more sensitive to OS than WT but that have a similar overall transcriptomic profile compared to the WT have been excluded from this analysis as their sensitivity to OS could be explained by LOF hits in protein-coding genes independently of expression (Methods). N = 3 and 4 for the WT-like sensitivity and more sensitive than WT categories, respectively. Middle of boxplot represents the median, boundary of boxes represents the 25th and 75th percentile (interquartile range), and whiskers represent the largest or smallest data point that is no farther than 1.5 times the interquartile range from box boundaries.

# Reporting Summary

## Statistics

For all statistical analyses, confirm that the following items are present in the figure legend, table legend, main text, or Methods section.

| n/a | Confirmed | |
|---|---|---|
| ☐ | ☒ | The exact sample size (*n*) for each experimental group/condition, given as a discrete number and unit of measurement |
| ☐ | ☒ | A statement on whether measurements were taken from distinct samples or whether the same sample was measured repeatedly |
| ☐ | ☒ | The statistical test(s) used AND whether they are one- or two-sided<br>*Only common tests should be described solely by name; describe more complex techniques in the Methods section.* |
| ☐ | ☒ | A description of all covariates tested |
| ☐ | ☒ | A description of any assumptions or corrections, such as tests of normality and adjustment for multiple comparisons |
| ☐ | ☒ | A full description of the statistical parameters including central tendency (e.g. means) or other basic estimates (e.g. regression coefficient) AND variation (e.g. standard deviation) or associated estimates of uncertainty (e.g. confidence intervals) |
| ☐ | ☒ | For null hypothesis testing, the test statistic (e.g. $F$, $t$, $r$) with confidence intervals, effect sizes, degrees of freedom and $P$ value noted<br>*Give P values as exact values whenever suitable.* |
| ☒ | ☐ | For Bayesian analysis, information on the choice of priors and Markov chain Monte Carlo settings |
| ☐ | ☒ | For hierarchical and complex designs, identification of the appropriate level for tests and full reporting of outcomes |
| ☐ | ☒ | Estimates of effect sizes (e.g. Cohen's *d*, Pearson's *r*), indicating how they were calculated |

*Our web collection on statistics for biologists contains articles on many of the points above.*

## Software and code

Policy information about availability of computer code

| Data collection | All available softwares used were described in the methods along with the version: Varscan2.4.6, fastp v 0.23.1, samtools v1.17, picard v2.26.3, subread v2.0.6, DESeq2 v1.34.0, drc v3.0-1, growthcurver v0.3.1, vegan (2.7-2), CytExpert (2.5) |
|---|---|
| Data analysis | The variant calling and RNA-seq analysis pipeline is available at https://github.com/arnaud00013/Experimental_evolution_S_pombe |

For manuscripts utilizing custom algorithms or software that are central to the research but not yet described in published literature, software must be made available to editors and reviewers. We strongly encourage code deposition in a community repository (e.g. GitHub). See the Nature Portfolio guidelines for submitting code & software for further information.

## Data

Policy information about availability of data

All manuscripts must include a data availability statement. This statement should provide the following information, where applicable:
- Accession codes, unique identifiers, or web links for publicly available datasets
- A description of any restrictions on data availability
- For clinical datasets or third party data, please ensure that the statement adheres to our policy

Raw DNA and RNA sequences are available on NCBI: https://www.ncbi.nlm.nih.gov/bioproject/PRJNA1315931. There are no restrictions on data availability.

# Research involving human participants, their data, or biological material

Policy information about studies with [human participants or human data](). See also policy information about [sex, gender (identity/presentation), and sexual orientation]() and [race, ethnicity and racism]().

| | |
|---|---|
| Reporting on sex and gender | Sex and gender are not relevant to this study as it studies yeast populations and not human populations. |
| Reporting on race, ethnicity, or other socially relevant groupings | Race, ethnicity, or other socially relevant groupings are not relevant to this study as it studies yeast populations and not human populations. |
| Population characteristics | See above. |
| Recruitment | See above. |
| Ethics oversight | See above. |

Note that full information on the approval of the study protocol must also be provided in the manuscript.

# Field-specific reporting

Please select the one below that is the best fit for your research. If you are not sure, read the appropriate sections before making your selection.

☒ Life sciences ☐ Behavioural & social sciences ☐ Ecological, evolutionary & environmental sciences

For a reference copy of the document with all sections, see [nature.com/documents/nr-reporting-summary-flat.pdf]()

# Life sciences study design

All studies must disclose on these points even when the disclosure is negative.

| | |
|---|---|
| Sample size | Sample size was determined based on the standards in the field. The Lenski LTEE had 12 replicates, which has been found to be sufficient to analyze several aspects of evolution. |
| Data exclusions | We excluded 5 S. pombe populations from analysis as they were contaminated or lost after 10,000 generations of evolution. |
| Replication | The reported study is 10 parallel populations as replication of the evolution. For DNA analysis, time-resolved sequencing allows intrinsic replication of discovered novel alleles. Phenotyping and RNA-seq was done as biological replications over different days. |
| Randomization | The clonal ancestors were chosen randomly from single colonies from a plate. |
| Blinding | Blinding to the experimental environment was not relevant to our study because the ten populations were subjected to identical data collection and data analysis procedures. |

# Reporting for specific materials, systems and methods

We require information from authors about some types of materials, experimental systems and methods used in many studies. Here, indicate whether each material, system or method listed is relevant to your study. If you are not sure if a list item applies to your research, read the appropriate section before selecting a response.

## Materials & experimental systems

| n/a | Involved in the study |
|---|---|
| ☒ ☐ | Antibodies |
| ☒ ☐ | Eukaryotic cell lines |
| ☒ ☐ | Palaeontology and archaeology |
| ☒ ☐ | Animals and other organisms |
| ☒ ☐ | Clinical data |
| ☒ ☐ | Dual use research of concern |
| ☒ ☐ | Plants |

## Methods

| n/a | Involved in the study |
|---|---|
| ☒ ☐ | ChIP-seq |
| ☒ ☐ | Flow cytometry |
| ☒ ☐ | MRI-based neuroimaging |

## Plants

Seed stocks

No plants were used in this study.

Novel plant genotypes

No plants were used in this study.

Authentication

No plants were used in this study.

