## [Peer Review File · Nature Ecology & Evolution]

Parallel but distinct adaptive routes in the budding and fission yeasts after 10,000 generations of experimental evolution

Corresponding Author: Professor Alex Nguyen Ba

Version 0:

Decision Letter:

19th November 2025

Dear Alex,

Your manuscript entitled "Parallel but distinct adaptive routes in the budding and fission yeasts after 10,000 generations of experimental evolution" has now been seen by three experts in yeast evolutionary genomics and experimental evolution. The reviewers have raised a number of concerns which will need to be addressed before we can offer publication in Nature Ecology & Evolution. We will therefore need to see your responses to the criticisms raised and to some editorial concerns, along with a revised manuscript, before we can reach a final decision regarding publication.

We therefore invite you to revise your manuscript taking into account all reviewer and editor comments. Please highlight all changes in the manuscript text file in Microsoft Word format.

* If you have not done so already please begin to revise your manuscript so that it conforms to our Article format instructions at <http://www.nature.com/natecolevol/info/final-submission>. Refer also to any guidelines provided in this letter.

* Extended Data Figures - please ensure that any supplementary figures and tables that are crucial to the manuscript's conclusions are converted into Extended Data figures and tables to increase visibility of these data. Extended Data figures and tables are online-only (present in the online PDF and full-text HTML versions of the paper), peer-reviewed display items that provide essential background to the article but are not included in the main article due to space constraints. A maximum of ten Extended Data display items (figures and tables) is permitted.

Link Redacted

We hope to receive your revised manuscript within four to eight weeks. If you cannot send it within this time, please let us

know. We will be happy to consider your revision so long as nothing similar has been accepted for publication at Nature Ecology & Evolution or published elsewhere.

Nature Ecology & Evolution is committed to improving transparency in authorship. As part of our efforts in this direction, we are now requesting that all authors identified as 'corresponding author' on published papers create and link their Open Researcher and Contributor Identifier (ORCID) with their account on the Manuscript Tracking System (MTS), prior to acceptance. ORCID helps the scientific community achieve unambiguous attribution of all scholarly contributions. You can create and link your ORCID from the home page of the MTS by clicking on 'Modify my Springer Nature account'. For more information please visit www.springernature.com/orcid.

[redacted]

Reviewers' comments:

Reviewer #1 (Remarks to the Author):

The paper compares an experimental evolution approach in *S. pombe* (Sp) with previously published work in *S. cerevisiae* (Sc) in a similar set up. The authors observed both parallelism as well as species uniqueness in the evolutionary dynamic. The study is interesting and well presented, although I feel the results obtained are perhaps expected and remain descriptive. Moreover, a direct comparison between the two fundamentally different biological systems is technically difficult.

- "This slower rate of fitness gain is not due to *S. pombe* being 'pre-adapted' to this environment, as the ancestor *S. cerevisiae* strain used in a previous study is much fitter than the ancestor *S. pombe* strain used here. *S. pombe* adapt less compare to Sc in the same conditions and is not due to initial fitness." One possible bias of this experimental design is the use of YPD as base media - which is not the ideal media for pombe (which is growth in YES) . So Sp has to adapt not only to the hypoxic condition but also to a suboptimal media, and these conditions might have a synergistic effect. Similarly, Sp growth faster at 32 in YES. The combination of these factors might affect the downstream outcome of the selection experiment, the two species are not exposed to the same selection, so perhaps is normal to observe quantitative differences in accumulation of new mutations and different target genes.

- "We used a flow cytometer to measure the frequency of the dark *S. pombe* populations and the green *S. cerevisiae* strain daily, and assessed fitness based on changes in frequency (Equation 1 in Methods)." What is the rationale of measuring fitness changes of the Sp lines using the Sc and not the Sp ancestor? Can you rule out any type of interactions between the species that could differentially affect the evolved Sp clones? The logic should be explained in the text.

- Is it possible to quantify the parallelism within and between the Sc and Sp systems (e.g. how many parallel and different solutions are there within and between systems)?

- Did the previous study have shown a role for aneuploidies in Sc adaptation? Aneuploidies emerge often in Sc experimental evolution whereas are likely inviable in Sp - this could contribute to the overall adaptation dynamic. Similarly, are all the lines remaining haploids in both species?

- Page and line numbers should have been added - will help reviewers to direct comments

Reviewer #2 (Remarks to the Author):

N'Guessan et al., Parallel but distinct adaptive routes in the budding and fission yeasts after 10,000 generations of experimental evolution

Summary

This is a very interesting study that compares long-term experimental evolution in two distantly related yeast species, *Saccharomyces cerevisiae* (budding yeast) and *Schizosaccharomyces pombe* (fission yeast), after 10,000 generations under similar, high-sugar, hypoxic conditions.

Both species adapt to the growth environment, but *S. pombe* populations show significantly slower fitness gains than *S. cerevisiae*, a difference that was not attributed to initial pre-adaptation. *S. pombe* populations displayed predominantly fix mutations in non-coding regions, with a bias for insertions (indels), whereas *S. cerevisiae* populations tend to fix more loss-of-function (LOF) mutations in coding regions. The mutation patterns are only partially explained by differences in genome architecture, suggesting selection played a significant role.

Most evolved *S. pombe* populations show increased sensitivity to oxidative stress, which was not observed in *S. cerevisiae*. These trade-offs are driven by mutations that downregulate respiration, shifting reliance to fermentation, which increases reactive oxygen species (ROS). Adaptation in *S. pombe* involves upregulation of heme metabolism, transporters, and autophagy pathways, mostly through transcriptional changes rather than only coding sequence mutations. Many of these changes appear to be due to trans-regulatory effects rather than cis-acting mutations.

There is evolutionary parallelism because both organisms display adaptation to hypoxic stress through increased fermentation and altered stress responses, but the molecular mechanisms differ. Ultimately, the study underscores the role of contingency, i.e. past evolutionary events and genetic architecture, in shaping adaptation, addressing how closely related species can evolve similar phenotypes through distinct genetic mechanisms.

General Comments

The authors are highly experienced in this field of study, and the experimental methodology is relatively straightforward, and thus, while I have not gone through the data analysis in detail, I assume it is perfectly reasonable and the raw sequence data has been made available. *S. cerevisiae* and *S. pombe* are the most well studied yeast species, with well-established research communities dedicated to each organism, and this work will be of interest to both groups. The results are important and novel because this type of detailed comparison between the two species has never been done before. The current literature is represented well, and the discussion of the results is interesting.

I found it particularly interesting that “The fission yeast populations mainly evolved through mutations in non-coding regions with a bias toward fixing indels and not fixing coding regions SNVs (Figure2B), which is different than what has been reported for the budding yeast.” I wonder if this has something to do with the molecular source of the background mutations in *S. pombe* vs *S. cerevisiae*, which occur at similar rates but may have a different molecular basis. Perhaps this could be analyzed further in the context of DNA repair pathways.

Specific Comments

The methods could be expanded to clearly represent source of strains, genotypes, and growth conditions applied.

e.g. The genotype of the *S. pombe* strain should be included. The genotype of the *S. cerevisiae* reference strain expressing GFP is unusual: Commas are not usually included after marker genes, D= delta, *S. pombe* genes are represented in lowercase letters.

(YAN438: MAT α , his3D1, ura3D0, leu2D0, lys2D0, can1::RPL39pr_ymGFP_Ste2pr-SpHIS5_Ste3pr_LEU2) for 24 hours

Reviewer #3 (Remarks to the Author):

SUMMARY

I have reviewed the manuscript “Parallel but distinct adaptive routes in the budding and fission yeasts after 10,000 generations of experimental evolution” by N’Guessan et al. The work addresses fundamental questions about contingency in evolutionary, namely to what extent pre-existing differences between species alters their adaptation to a given environment. The authors evolved 10 populations of *S. pombe* for 10,000 generations in identical conditions to those used for 10,000 generations of *S. cerevisiae*. They find that *pombe* adapts more slowly than *cerevisiae* to this condition. From sequencing these populations and identifying likely targets of positive selection, they find several differences in the evolution of *pombe* and *cerevisiae*. In particular, they report a difference in the spectrum of mutations acquired and a difference in the identity of the genes that were putative targets of positive selection. They also find that 7 of the populations have acquired increased sensitivity to oxidative stress as a result of adaptation to this condition, which the authors associate with transcriptional changes in the populations.

I generally found the manuscript to be convincing and the differences between the evolution of the two species in the same condition to be very interesting. I am particularly excited by the notion that differences in the life history and base metabolism of these species drives differences in their adaptation to this condition and their propensity to tradeoffs with oxidative stress. That being said, I think there are a few places the manuscript could be improved, which I detail below, in no particular order.

POINTS TO ADDRESS

While I am generally convinced that the authors did everything they could to keep the environmental conditions between the *cerevisiae* and *pombe* evolution experiments consistent, I wonder how much differences in the baseline growth rates of the two species create differences in the conditions and thus drive difference between the evolution of the species. In particular, the methods for the fitness measurement experiment indicate that *cerevisiae* has a much higher growth rate in this condition. This likely shortens the amount of time *cerevisiae* spends actively fermenting compared to *pombe* in this condition. Is this a concern? I think a discussion of these

“petite-positive” - I think this could use a definition/reminder for non-yeast readers.

I found “In protein-coding genes, most indels are either short (1 or 2 bp) due to their higher likelihood (Supplementary Figure 3) or do not cause a frameshift in the gene, in contrast to indels in non-coding regions” to be a bit confusing on first read through. Since the point of this sentence (I think) is that frameshift are under purifying selection, I might rephrase to exclude the piece about 1-2bp and focus on the enrichment of those that prevent frameshift.

I found the difference in the number of non-coding vs coding hits in *s. pombe* compared to *s. cerevisiae* to be very interesting. The authors provide some data that suggests that there is increased purifying selection on coding regions in *s. pombe*. I think a bit more discussion about the underlying differences between the species that drive this would be of

interest. In particular, one implicit hypothesis is that, perhaps, the whole genome duplication in *s. cerevisiae* has created more redundancy in the genome that can be tinkered with via coding mutations. The authors hint at this possibility when noting that lack of IRA1/IRA2 (*gap1*) mutations in the discussion, but I would appreciate a more explicit discussion of these ideas and may also help inform how much the roles of genomic context vs. life history drive the differences between these species.

Supplementary Figure 1 - why are there 11 points in *s. pombe*? I assumed this was each population but only 10/15 made it to 10,000 generations?

Supplementary Figure 1 - Are the 3 populations with the lowest fitness gain the same as those with the least amount of oxidative stress? It would be interesting to see the relationship to strengthen the claim of a tradeoff between adaptation in the environment and stress. Alternatively, is there anything in the mutations and/or transcriptional profiles of the other populations to indicate how they avoid this tradeoff?

It wasn't immediately clear to me that non-coding cis-regulatory regions were included in the multi-hit genes analysis. I think this should be explicitly stated in the main text.

I found a couple of the paragraphs detailing mutation information to be long and a bit dense,. In particular, "Most of the *S. pombe* multi-hit genes..." in the results and "A third pathway frequently mutated..." in the discussion. In my view, these detail some of the most interesting results of the paper, but it took some time for me to parse them. Consider splitting these into multiple paragraphs with a bit more signposting.

Table 1 - Table 1 could be improved to include the kinds of mutations in/near each of these genes rather than just listing LOF information. Are those where it says "Not Found" often regulatory mutations? Putative gain or modification of function? Is the gene a positive or negative regulator of the process of interest? I think this would help with interpretation of the mutation results. I also think a reorganization of the table to include categories of genes ("HO-related", "OS-related", "other", etc) would help with the parsing of the genes and relationship with the text that I mention above.

It was not immediately clear from the text nor methods which samples were selected for gene expression analysis. Because only broad comparisons were conducted between the evolved strains and wild type, it initially seemed like only a single sample was measured, but upon looking at the Github, it appears that all populations were sequenced, with only the tradeoff populations used for the comparison. A short description of the procedure at the beginning of this results section and in the methods is warranted.

Additionally, given the authors have gene expression data from all the populations, it would be of interest to show details on variability between the populations with regards to gene expression. In particular, I'm curious how many populations down-regulated *pyk1*, and if *pyk1*'s down-regulation is exclusive to those with mutations in/near the gene, suggesting it is driven primarily by mutations in cis-regulatory regions, or if there may be evidence for trans-regulatory effects in the other populations.

*****END*****

Version 1:

Decision Letter:

8th January 2026

Dear Alex,

Thank you for submitting your revised manuscript "Parallel but distinct adaptive routes in the budding and fission yeasts after 10,000 generations of experimental evolution" (NATECOLEVOL-25093234A). It has now been seen again by two of the original reviewers and their comments are below. The reviewers find that the paper has improved in revision, and therefore we'll be happy in principle to publish it in *Nature Ecology & Evolution*, pending minor revisions to comply with our editorial and formatting guidelines.

Thank you again for your interest in *Nature Ecology & Evolution*. Please do not hesitate to contact me if you have any questions.

[redacted]

Reviewer #1 (Remarks to the Author):

Thank you to the authors to reply to my comments, I do not have further remarks.

Reviewer #3 (Remarks to the Author):

The authors have addressed my comments, and I think the manuscript is much improved. I have no further concerns that need to be addressed.

Reviewer #1 (Remarks to the Author):

The paper compares an experimental evolution approach in *S. pombe* (Sp) with previously published work in *S. cerevisiae* (Sc) in a similar set up. The authors observed both parallelism as well as species uniqueness in the evolutionary dynamic. The study is interesting and well presented, although I feel the results obtained are perhaps expected and remain descriptive. Moreover, a direct comparison between the two fundamentally different biological systems is technically difficult.

A: We thank the reviewer for these comments and have added some of these to the text to acknowledge the strength and weaknesses of our manuscript.

- “This slower rate of fitness gain is not due to *S. pombe* being ‘pre-adapted’ to this environment, as the ancestor *S. cerevisiae* strain used in a previous study is much fitter than the ancestor *S. pombe* strain used here. *S. pombe* adapt less compare to Sc in the same conditions and is not due to initial fitness.” One possible bias of this experimental design is the use of YPD as base media - which is not the ideal media for pombe (which is growth in YES) . So Sp has to adapt not only to the hypoxic condition but also to a suboptimal media, and these conditions might have a synergistic effect. Similarly, Sp growth faster at 32 in YES. The combination of these factors might affect the downstream outcome of the selection experiment, the two species are not exposed to the same selection, so perhaps is normal to observe quantitative differences in accumulation of new mutations and different target genes.

A: We also agree that the same environment does not necessarily indicate the same selection pressure on cells. In our original manuscript, we discussed this when comparing the different genes mutated between the two species.

Furthermore, we emphasize that the lower rate of adaptation of *S. pombe* in this experiment is probably not due to some fundamental aspect of *S. pombe*, though some of the general ideas behind evolvability due to past whole-genome duplication events or due to other life history traits might contribute to this effect. In other environments, for example YES media, it is possible that *S. pombe* may evolve faster than *S. cerevisiae*. Throughout the text, we have made it clearer that the observations were made in YPD or “in our growth conditions”. We have also added more discussion of these caveats more clearly in the discussion: “We note, however, that maintaining environmental conditions between species does not necessarily translate to identical selection pressures, and as such direct comparison between species and evolutionary forecasting with cross-species comparisons is still technically challenging. Despite this, our study suggests that [...]”

- “We used a flow cytometer to measure the frequency of the dark *S. pombe* populations and the

green *S. cerevisiae* strain daily, and assessed fitness based on changes in frequency (Equation 1 in Methods).” What is the rationale of measuring fitness changes of the *Sp* lines using the *Sc* and not the *Sp* ancestor? Can you rule out any type of interactions between the species that could differentially affect the evolved *Sp* clones? The logic should be explained in the text.

A: There is no compelling reason to use *S. cerevisiae* as a comparative strain for competitive fitness assays, other than it happened to be a strain that we knew did not have killer M1 virus (as opposed to the yeast strain of our previous study) and that contained a fluorescent protein that was very bright and had been validated in the past for flow cytometry.

Indeed, we attempted to make a fluorescent *S. pombe* strain for this as well (expressing mNeonGreen under the TetON promoter), but failed to produce a strain that was fluorescent enough. We do not think this is due to some fundamental limitation on fluorescent proteins in *S. pombe*, but simply that there are many things to try when making these constructs and we did not explore this in detail.

That being said, we did try to minimize species interaction during the competitive growth assay by putting the *S. cerevisiae* reference at ~10% of the population. We found approximately the same fitness measurements when *S. cerevisiae* was put at ~50%, but the measurements were very noisy and so cannot rule out species interaction (and more importantly, a change in species interaction over the 10,000 generations of evolution).

We explain both of these as our rationale in the methods section now, but we acknowledge that using an *Sp* ancestor strain might have been preferable. However, our experiment used other species as well (briefly mentioned but not analyzed), and our struggles with genetic modifications in some of these species was a major motivation for using *Sc*.

- Is it possible to quantify the parallelism within and between the *Sc* and *Sp* systems (e.g. how many parallel and different solutions are there within and between systems)?

A: We thank the reviewer for this suggestion. It is a very interesting analysis to do, though we note that we are particularly underpowered to do analysis (only 10 populations).

To do this analysis, we resampled the number of fixed mutations in our experiment from the list of gene hits in the experiments in the *S. cerevisiae* system (which had parallelism), treating each gene as equally likely to receive a mutation. Based on this sampling, we would have expected about 24 genes to be hit in 2 populations or more. The expectation is a bit higher than what we observed (19), however it is not statistically significant (p -value = 0.15, one-tailed CDF).

We also performed the same population hit parallelism analysis as in Johnson et al, allowing another comparison that shows that parallelism is indeed present in the *S. pombe* data (Fig S4).

One reason why the *S. pombe* system may have fewer parallel paths (though, we reiterate that this is not statistically significant) is that the *S. pombe* system adapted relatively slower

compared to *S. cerevisiae*, indicating that some of the major hits in the *S. cerevisiae* system are likely of strong effect. Indeed, the *ade2*- pressure is very strong in *S. cerevisiae*. We believe that if we had begun our *S. pombe* evolution with an *ade*- strain, we may have observed approximately the same amount of parallelism.

- Did the previous study have shown a role for aneuploidies in *Sc* adaptation? Aneuploidies emerge often in *Sc* experimental evolution whereas are likely inviable in *Sp* - this could contribute to the overall adaptation dynamic. Similarly, are all the lines remaining haploids in both species?

A: The previous study did not find evidence of aneuploidies in the evolution of *S. cerevisiae*. This has been traced to the fact that the previous study's strain contains a loss-of-function mutation in the gene *ssd1*, which seems to prevent the rise of large-scale aneuploidies (see Tung, G3, 2021).

Aneuploidies are readily identifiable during time-course allele frequency analyses as mutations fix at some characteristic fractions ($1/2$ or $1/3$ etc, see Rego-Costa, G3, 2023). We did not find evidence of large-scale aneuploidies in our *S. pombe* populations during the analysis of sequencing data.

- Page and line numbers should have been added - will help reviewers to direct comments

A: We have added this.

Reviewer #2 (Remarks to the Author):

N'Guessan et al., Parallel but distinct adaptive routes in the budding and fission yeasts after 10,000 generations of experimental evolution

Summary

This is a very interesting study that compares long-term experimental evolution in two distantly related yeast species, *Saccharomyces cerevisiae* (budding yeast) and *Schizosaccharomyces pombe* (fission yeast), after 10,000 generations under similar, high-sugar, hypoxic conditions.

Both species adapt to the growth environment, but *S. pombe* populations show significantly slower fitness gains than *S. cerevisiae*, a difference that was not attributed to initial pre-adaptation. *S. pombe* populations displayed predominantly fix mutations in non-coding regions, with a bias for insertions (indels), whereas *S. cerevisiae* populations tend to fix more loss-of-function (LOF) mutations in coding regions. The mutation patterns are only partially explained by differences in genome architecture, suggesting selection played a significant role.

Most evolved *S. pombe* populations show increased sensitivity to oxidative stress, which was not observed in *S. cerevisiae*. These trade-offs are driven by mutations that downregulate respiration, shifting reliance to fermentation, which increases reactive oxygen species (ROS). Adaptation in *S. pombe* involves upregulation of heme metabolism, transporters, and autophagy pathways, mostly through transcriptional changes rather than only coding sequence mutations. Many of these changes appear to be due to trans-regulatory effects rather than cis-acting mutations.

There is evolutionary parallelism because both organisms display adaptation to hypoxic stress through increased fermentation and altered stress responses, but the molecular mechanisms differ. Ultimately, the study underscores the role of contingency, i.e. past evolutionary events and genetic architecture, in shaping adaptation, addressing how closely related species can evolve similar phenotypes through distinct genetic mechanisms.

General Comments

The authors are highly experienced in this field of study, and the experimental methodology is relatively straightforward, and thus, while I have not gone through the data analysis in detail, I assume it is perfectly reasonable and the raw sequence data has been made available. *S. cerevisiae* and *S. pombe* are the most well studied yeast species, with well-established research communities dedicated to each organism, and this work will be of interest to both groups. The results are important and novel because this type of detailed comparison between the two species has never been done before. The current literature is represented well, and the discussion of the results is interesting.

A: We thank the reviewer for these comments.

I found it particularly interesting that “The fission yeast populations mainly evolved through mutations in non-coding regions with a bias toward fixing indels and not fixing coding regions SNVs (Figure2B), which is different than what has been reported for the budding yeast.” I wonder if this has something to do with the molecular source of the background mutations in *S. pombe* vs *S. cerevisiae*, which occur at similar rates but may have a different molecular basis. Perhaps this could be analyzed further in the context of DNA repair pathways.

A: We discussed this briefly in the text. Indeed, the spectrum of mutation from mutational accumulation studies indicates that indels are more common than substitutions in *S. pombe* (Behringer and Hall, G3, 2015), which presumably is due to changes in DNA repair pathways. If indels are the main driver of molecular evolution, and if there is strong purifying selection on gene function because of the lack of genetic redundancy (or other), then a predominance of mutations in non-coding regions is exactly what one would expect. *S. pombe* also has a much higher essential gene load than *S. cerevisiae* (26.1% vs 17.8%, Kim et al, Nature Biotechnology, 2010), which may be another reason for the signature of purifying selection we observe.

Specific Comments

The methods could be expanded to clearly represent source of strains, genotypes, and growth conditions applied.

e.g. The genotype of the *S. pombe* strain should be included. The genotype of the *S. cerevisiae* reference strain expressing GFP is unusual: Commas are not usually included after marker genes, D= delta, *S. pombe* genes are represented in lowercase letters.

(YAN438: MAT α , his3D1, ura3D0, leu2D0, lys2D0, can1::RPL39pr_ymGFP_Ste2pr-SpHIS5_Ste3pr_LEU2) for 24 hours

A: We included the *S. pombe* strain (972h-), which is a standard, reference laboratory strain of *S. pombe*. We fixed the genotype description for the *S. cerevisiae* strain expressing GFP. The nomenclature we used is due to limitations of our strain management database.

We also describe the growth conditions applied and the methods section now briefly includes: “We evolved 15 *S. pombe* populations of the laboratory strain 972h- at 30 °C in wells of a lidded, flat-bottom 96-well plate containing 128 μ l of YPD plus antibiotics (1% yeast extract, 2% peptone, 2% dextrose, 100 μ g/mL ampicillin, 25 μ g/mL tetracycline) for 10,000 generations, collecting and freezing samples at -80 °C every 70 generations.”

Reviewer #3 (Remarks to the Author):

SUMMARY

I have reviewed the manuscript “Parallel but distinct adaptive routes in the budding and fission yeasts after 10,000 generations of experimental evolution” by N’Guessan et al. The work addresses fundamental questions about contingency in evolutionary, namely to what extent pre-existing differences between species alters their adaptation to a given environment. The authors evolved 10 populations of *S. pombe* for 10,000 generations in identical conditions to those used for 10,000 generations of *S. cerevisiae*. They find that *pombe* adapts more slowly than *cerevisiae* to this condition. From sequencing these populations and identifying likely targets of positive selection, they find several differences in the evolution of *pombe* and *cerevisiae*. In particular, they report a difference in the spectrum of mutations acquired and a difference in the identity of the genes that were putative targets of positive selection. They also find that 7 of the populations have acquired increased sensitivity to oxidative stress as a result of adaptation to this condition, which the authors associate with transcriptional changes in the populations.

I generally found the manuscript to be convincing and the differences between the evolution of the two species in the same condition to be very interesting. I am particularly excited by the notion that differences in the life history and base metabolism of these species drives differences

in their adaptation to this condition and their propensity to tradeoffs with oxidative stress. That being said, I think there are a few places the manuscript could be improved, which I detail below, in no particular order.

A: We thank the reviewer for these comments.

POINTS TO ADDRESS

While I am generally convinced that the authors did everything they could to keep the environmental conditions between the cerevisiae and pombe evolution experiments consistent, I wonder how much differences in the baseline growth rates of the two species create differences in the conditions and thus drive difference between the evolution of the species. In particular, the methods for the fitness measurement experiment indicate that cerevisiae has a much higher growth rate in this condition. This likely shortens the amount of time cerevisiae spends actively fermenting compared to pombe in this condition. Is this a concern? I think a discussion of these

A: We now emphasize that it is only the growing condition that is identical between the S. pombe and S. cerevisiae experiments (they were passaged at the same time using liquid handling robotics). Furthermore, we now make it clear that the selection pressure on these species is likely different due to genetic context, growth rates etc.

Generally, we do not know if there is a method to very precisely control for the selection pressure faced by the cells and it would be interesting to redo this experiment in light of such approach if it existed.

As for the amount of time in fermentation vs stationary, there is indeed substantial differences between the two species. We also agree that this can in principle alter selection targets for the two species and over the course of the evolution experiment as strains increase in fitness and increase the duration of stationary phase over long periods of time. Since we cannot control for this easily (at least not anymore), we have added a small amount of discussion to discuss this caveat in our experimental design: “We note, however, that maintaining environmental conditions between species does not necessarily translate to identical selection pressures, and as such direct comparison between species and evolutionary forecasting with cross-species comparisons is still technically challenging.”

“petite-positive” - I think this could use a definition/reminder for non-yeast readers.

A: We have added a sentence to define this term to emphasize the reliance of S. pombe on oxidative phosphorylation.

I found “In protein-coding genes, most indels are either short (1 or 2 bp) due to their higher likelihood (Supplementary Figure 3) or do not cause a frameshift in the gene, in contrast to indels in non-coding regions” to be a bit confusing on first read through. Since the point of this

sentence (I think) is that frameshift are under purifying selection, I might rephrase to exclude the piece about 1-2bp and focus on the enrichment of those that prevent frameshift.

A: We thank the reviewer for the suggestion in improving the clarity of this sentence and have reworded it: “In protein-coding genes, controlling for increased likelihood of shorter indels, we find that indels that do not cause frameshifts are enriched compared to the ones that do (Supplementary Fig. 3). In contrast, indels in non-coding regions do not exhibit such selective constraints on their length.”

I found the difference in the number of non-coding vs coding hits in *s. pombe* compared to *s. cerevisiae* to be very interesting. The authors provide some data that suggests that there is increased purifying selection on coding regions in *s. pombe*. I think a bit more discussion about the underlying differences between the species that drive this would be of interest. In particular, one implicit hypothesis is that, perhaps, the whole genome duplication in *s. cerevisiae* has created more redundancy in the genome that can be tinkered with via coding mutations. The authors hint at this possibility when noting that lack of IRA1/IRA2 (*gap1*) mutations in the discussion, but I would appreciate a more explicit discussion of these ideas and may also help inform how much the roles of genomic context vs. life history drive the differences between these species.

A: We have added a sentence in the discussion of the results regarding this possible hypothesis: “This and the paucity of fixed coding mutations compared to *S. cerevisiae* populations suggest that the genetic context of *S. pombe* is less amenable to tinkering through loss-of-function mutations. One hypothesis for this may be that the whole-genome duplication event in *S. cerevisiae* provided more genetic redundancy, which has been suggested to increase evolvability through coding mutations or through rapid changes in functional gene content (see Discussion).”

We acknowledge, however, that this is simply a hypothesis and that differences in selection pressure may yield different outcomes.

Supplementary Figure 1 - why are there 11 points in *s. pombe*? I assumed this was each population but only 10/15 made it to 10,000 generations?

A: We thank the reviewer for pointing that out. This is an artifact from our plotting method. We added a jitter plot over the boxplot so the outlier is represented twice here (boxplots sometimes plot outliers as individual points). We adjusted this setting for both distributions.

Supplementary Figure 1 - Are the 3 populations with the lowest fitness gain the same as those with the least amount of oxidative stress? It would be interesting to see the relationship to strengthen the claim of a tradeoff between adaptation in the environment and stress.

Alternatively, is there anything in the mutations and/or transcriptional profiles of the other populations to indicate how they avoid this tradeoff?

A: We performed several lines of analyses related to these questions but unfortunately have been limited by statistical power (only 10 populations) and there were no statistically significant differences in these comparisons. That being said, the trends are consistent overall with the rest of the analyses. For example, the three populations with less fitness gain have higher average *pyk1* expression levels (but not statistically significant through a more robust Wilcoxon rank sum test). The populations that were more resistant to oxidative stress (or as resistant to WT) seemed to also have higher *pyk1* expression level. Therefore, the tradeoff claim is still qualitative, i.e. most of the populations that adapted to hypoxic stress became more sensitive to oxidative stress (7 populations out of 10).

However, the challenge here is that oxidative stress sensitivity is not completely controlled by *pyk1* levels or that the shape of the tradeoff is not strictly linear dependent on the genes targeted.

We have toned down the statement that adaptation in our growing environment leads to oxidative stress sensitivity to make it clearer that this is not a necessary fact/tradeoff, but that we observed lower oxidative stress resistance for “most of our population”, which we suggest is due to hypoxic stress adaptation. That would indicate that *S. pombe* can still adapt in the presence of both hypoxic stress and oxidative stress (and some populations have indeed adapted without increased oxidative stress sensitivity – and those retain WT-like *pyk1* expression levels), but this is still speculative as there are many possible stresses in our environment and not only hypoxic stress. A more thorough exploration of this tradeoff likely requires experimental evolution in normoxic environment.

It wasn't immediately clear to me that non-coding cis-regulatory regions were included in the multi-hit genes analysis. I think this should be explicitly stated in the main text.

A: We have added a clarifying statement for what we consider multi-hit genes: “we identified 19 multi-hit genes with hits in either the cis-regulatory region or the coding region”

I found a couple of the paragraphs detailing mutation information to be long and a bit dense. In particular, “Most of the *S. pombe* multi-hit genes...” in the results and “A third pathway frequently mutated...” in the discussion. In my view, these detail some of the most interesting results of the paper, but it took some time for me to parse them. Consider splitting these into multiple paragraphs with a bit more signposting.

A: We have rephrased and split these paragraphs for added clarity.

Table 1- Table 1 could be improved to include the kinds of mutations in/near each of these genes rather than just listing LOF information. Are those where it says “Not Found” often regulatory mutations? Putative gain or modification of function? Is the gene a positive or negative regulator

of the process of interest? I think this would help with interpretation of the mutation results. I also think a reorganization of the table to include categories of genes (“HO-related”, “OS-related”, “other”, etc) would help with the parsing of the genes and relationship with the text that I mention above.

A: We updated Table 1 to include the requested data when such information was available in the literature and database annotations.

It was not immediately clear from the text nor methods which samples were selected for gene expression analysis. Because only broad comparisons were conducted between the evolved strains and wild type, it initially seemed like only a single sample was measured, but upon looking at the Github, it appears that all populations were sequenced, with only the tradeoff populations used for the comparison. A short description of the procedure at the beginning of this results section and in the methods is warranted.

A: We have added a clarifying statement for what exactly we use for the transcriptomic analysis: “To do so, we performed RNA-seq of the WT ancestor and of the final evolved populations and compared transcriptomic profiles of populations more sensitive to H₂O₂ than WT to populations as sensitive as WT.”.

We have also added detailed discussion in the methods/results section related to the analysis we performed. Briefly, after RNA-seq of all the populations and FPKM conversion, we performed PCA clustering (first 2 PCs had about 75% of the variance) and identified 2 main clusters (1 containing the WT). One of the clusters only contained populations that were more sensitive to oxidative stress. We then compared both clusters to each other for differential gene expression analysis, and found 9 of the 19 multi-hit genes to be differentially expressed. Important genes such as *pyk1* were found to be downregulated in those populations (Fig S6). The cumulative set of genes gave the figure included in our manuscript.

Additionally, given the authors have gene expression data from all the populations, it would be of interest to show details on variability between the populations with regards to gene expression. In particular, I’m curious how many populations down-regulated *pyk1*, and if *pyk1*’s down-regulation is exclusive to those with mutations in/near the gene, suggesting it is driven primarily by mutations in cis-regulatory regions, or if there may be evidence for trans-regulatory effects in the other populations.

A: We thank the reviewer for these suggestions. Therefore, we added the following sentences to the discussion: "In our dataset, 3 populations in which *pyk1* is downregulated have cis-regulatory mutations around the gene, while G2 is the only population in which *pyk1* is downregulated without such mutations (Supplementary Figures 5 and 6). This suggests that *pyk1*

downregulation can evolve more easily through cis-regulatory effects, but it can also evolve through trans-regulatory effects. "

We note that we failed to get sufficiently high-quality RNA for sequencing for H12 (This is now clear in the methods section), and H11 has a mutation in the coding sequencing of *pyk1*, which may be an alternative way to reduce Pyk1 activity without altering expression levels. H11 is more sensitive to oxidative stress than WT, even though it appears 'WT-like' by the transcriptome.

Reviewer #1:

Thank you to the authors to reply to my comments, I do not have further remarks.

A: We thank the reviewer for all comments throughout the publication of this manuscript.

Reviewer #3:

The authors have addressed my comments, and I think the manuscript is much improved. I have no further concerns that need to be addressed.

A: We thank the reviewer for all comments throughout the publication of this manuscript.